# Bring Your Own Algorithm for Optimal Differentially Private Stochastic Minimax Optimization

**Liang Zhang**
ETH Zurich & Max Planck Institute
liang.zhang@inf.ethz.ch

**Kiran Koshy Thekumparampil**
Amazon Search & UIUC
thekump2@illinois.edu

**Sewoong Oh**
University of Washington
sewoong@cs.washington.edu

**Niao He**
ETH Zurich
niao.he@inf.ethz.ch

## Abstract

We study *differentially private* (DP) algorithms for smooth stochastic minimax optimization, with stochastic minimization as a byproduct. The holy grail of these settings is to guarantee the optimal trade-off between the privacy and the excess population loss, using an algorithm with a *linear time-complexity* in the number of training samples. We provide a general framework for solving differentially private stochastic minimax optimization (DP-SMO) problems, which enables the practitioners to bring their own base optimization algorithm and use it as a black-box to obtain the near-optimal privacy-loss trade-off. Our framework is inspired from the recently proposed Phased-ERM method [22] for nonsmooth differentially private stochastic convex optimization (DP-SCO), which exploits the stability of the empirical risk minimization (ERM) for the privacy guarantee. The flexibility of our approach enables us to sidestep the requirement that the base algorithm needs to have bounded sensitivity, and allows the use of sophisticated variance-reduced accelerated methods to achieve near-linear time-complexity. To the best of our knowledge, these are the first near-linear time algorithms with near-optimal guarantees on the population duality gap for smooth DP-SMO, when the objective is (strongly-)convex–(strongly-)concave. Additionally, based on our flexible framework, we enrich the family of near-linear time algorithms for smooth DP-SCO with the near-optimal privacy-loss trade-off.

## 1 Introduction

Machine learning models are nowadays trained using large corpora of data samples collected from many different entities, e.g., from users of a large software service [28]. However, it has been empirically shown that these models can be exploited to reveal private information about these contributing entities. For example, Carlini et al. [13] attacked the large language model, GPT-2, to reveal hundreds of verbatim text samples used to train these models. These attacks violate the privacy of the contributing entities, but naturally they expect that no private information about them can be revealed through these models. Over the last decade, this expectation was even legislated into laws such as GDPR in EU [15]. There are many mathematical frameworks formalizing this expectation of privacy, but the most widely accepted one is that of *Differential Privacy* (DP) [18]. With high probability, models satisfying DP cannot be attacked by any adversary to identify that a particular training sample was used in its training. Hence, DP provides any entity plausible deniability that they contributed to the training set. However, optimization algorithms for training such models under DP require careful design choices to ensure privacy while preventing the degradation of convergence

36th Conference on Neural Information Processing Systems (NeurIPS 2022).

speed and data efficiency. This has led to the burgeoning field of differentially private optimization algorithms [6, 7, 22], which considers stochastic convex minimization as the canonical problem.

For differentially private stochastic convex optimization (DP-SCO) with $(\varepsilon, \delta)$-DP guarantees, the optimal excess population risk is $\Theta(1/(\mu n) + d \log(1/\delta)/(\mu n^2 \varepsilon^2))$ for $\mu$-strongly convex objectives, where $n$ is the number of participants and $d$ is the dimension of the variable. If the objective is also smooth, this can be achieved with a linear-time $(\varepsilon, \delta)$-DP algorithm [22] using $\mathcal{O}(n)$ stochastic gradient evaluations. This analysis critically relies on the concept of algorithmic *stability* [12], which measures how much the population loss of an algorithm's output changes when a single data point in the input dataset is perturbed. This is also known as *sensitivity* of an algorithm in DP, which determines how much noise needs to be added in order to achieve $(\varepsilon, \delta)$-DP (see Definition 2). We will use stability and sensitivity interchangeably when referring to optimization algorithms. As the sensitivity of stochastic gradient descent (SGD) is known [24], Feldman et al. [22] were able to add an appropriate amount of noise—tailored to this sensitivity—to an SGD-based algorithm to achieve optimal risk in linear time. However, such tight analysis of sensitivity is generally intractable for more complex optimization routines that practitioners might want to use. Further, the stability analysis of SGD only holds when the smoothness parameter $\ell$ of the problem is upper-bounded by $\tilde{\mathcal{O}}(\mu n)$. In the first part of the paper, we show that we can achieve similar guarantees using a wider choice of, potentially more practical, "base" algorithms, without any restrictions on the smoothness parameter.

On another front, many emerging practical machine learning applications are formulated as stochastic minimax optimization problems, e.g., generative adversarial networks [23], adversarially robust machine learning [35], and reinforcement learning [16]. Designing DP algorithms for solving these minimax problems is of paramount importance. For example, private generative adversarial networks provide a promising new direction to synthetic data generation [47], such as in networked time-series data [32]. Motivated by such applications, we study the differentially private stochastic minimax optimization (DP-SMO) problem of the form:

$$\min_{x \in \mathcal{X}} \max_{y \in \mathcal{Y}} F(x, y) \triangleq \mathbb{E}_\xi[f(x, y; \xi)],$$

where the objective $f(x, y; \xi)$ is smooth and convex-concave for any random vector $\xi$, and we are given access to $n$ i.i.d. samples $\{\xi_i\}_{i=1}^n$. In contrast to DP-SCO, where linear-time algorithms have been proposed to achieve optimal risk guarantees, existing private algorithms [11, 49] for DP-SMO achieving optimal guarantees have time-complexity which scales super-linearly in the number of samples $n$ (summarized in Table 1). In the second part of this paper, we close this gap by introducing a new class of output perturbation-based DP algorithms for both strongly-convex–strongly-concave and convex-concave settings, which can achieve near-optimal population risk bounds using $\tilde{\mathcal{O}}(n)$ stochastic gradient computations, where $\tilde{\mathcal{O}}$ hides logarithmic factors.

One of the main bottlenecks for (i) using a wider variety of algorithms to solve the DP-SCO problem, and (ii) solving the DP-SMO problem using linear-time algorithms, is the lack of known stability results for fast and sophisticated non-private algorithms, such as variance-reduced and accelerated methods. One can sidestep the algorithm-specific stability requirement by utilizing the stability of the optimal solution to a strongly-convex and Lipschitz empirical risk minimization (ERM) problem [42, 51]. Particularly, if the output of an algorithm for such an ERM problem is close enough to its empirical solution, then the sensitivity of the algorithm is automatically guaranteed. Exploiting this observation, Feldman et al. [22] proposed the phased-ERM algorithm for nonsmooth DP-SCO, which solves a series of strongly-convex ERM subproblems to sidestep the stability analysis of SGD for nonsmooth functions. This algorithm achieves a quadratic time-complexity for nonsmooth DP-SCO.

We observe that when the problem is additionally smooth, or when it allows a smooth minimax reformulation [37], there exist a plethora of fast algorithms to solve the resulting ERM subproblems. Combining these algorithms with (phased) output perturbation gives rise to a class of near-optimal near-linear [1] time-complexity private algorithms for both smooth DP-SCO and DP-SMO problems, without any additional effort on their stability analysis. Our contributions are summarized below:

- We introduce a flexible framework for solving smooth DP-SMO and smooth DP-SCO problems, utilizing the (phased) output perturbation mechanism. The black-box framework enables us to bypass the need to prove algorithm-specific stability and transform off-the-shelf optimization algorithms

---

[1] We only claim near-optimality of our rate because of its dependence on the condition number or additional logarithmic terms. We call an algorithm near-linear time when it is linear-time up to some logarithmic factors.

Table 1: Among $(\varepsilon, \delta)$-DP smooth minimax algorithms that achieve the near-optimal utility bound on the population strong duality gap for the $\mu$ strongly-convex–strongly-concave case and the population weak duality gap for the convex-concave case, the proposed framework achieves the best gradient complexity. Here $\tilde{\mathcal{O}}$ hides logarithmic terms, $\ell$ is the smoothness parameter, $\kappa = \ell/\mu$ and $d = \max\{d_x, d_y\}$. Lower-bounds are also summarized in the table.

| Settings | Lower-bound | Algorithm | Utility | Complexity |
|---|---|---|---|---|
| SC-SC | $\Omega\left(\frac{1}{\mu_x n} + \frac{d_x \log(1/\delta)}{\mu_x n^2 \varepsilon^2}\right)$ | Thm. 4.3 | $\mathcal{O}\left(\frac{\kappa^2}{\mu n} + \frac{\kappa^2 d \log(1/\delta)}{\mu n^2 \varepsilon^2}\right)$ | $\tilde{\mathcal{O}}(n + \sqrt{n}\kappa)$ |
| C-C | $\Omega\left(\frac{1}{\sqrt{n}} + \frac{\sqrt{d_x \log(1/\delta)}}{n\varepsilon}\right)$ | DP-SGDA [49] | $\mathcal{O}\left(\frac{1}{\sqrt{n}} + \frac{\sqrt{d \log(1/\delta)}}{n\varepsilon}\right)$ | $\mathcal{O}(n^{3/2}\sqrt{\varepsilon})$ |
| | | NSEG [11] | $\mathcal{O}\left(\frac{1}{\sqrt{n}} + \frac{\sqrt{d \log(1/\delta)}}{n\varepsilon}\right)$ | $\mathcal{O}(n^2)$ |
| | | NISPP [11] | $\mathcal{O}\left(\frac{1}{\sqrt{n}} + \frac{\sqrt{d \log(1/\delta)}}{n\varepsilon}\right)$ | $\tilde{\mathcal{O}}(n^{3/2})$ |
| | | Thm. 4.5 | $\tilde{\mathcal{O}}\left(\frac{1}{\sqrt{n}} + \frac{\sqrt{d \log(1/\delta)}}{n\varepsilon}\right)$ | $\tilde{\mathcal{O}}(n)$ |

into DP algorithms with near-optimal guarantees. This is attractive as there are currently no stability analyses for accelerated [2, 39] and variance-reduced [26, 39] algorithms for both SMO and SCO.

• Using the framework, we provide the first near-linear time private algorithms for smooth DP-SMO with near-optimal bound on the population duality gap, under both strongly-convex–strongly-concave and convex-concave cases (see Table 1). Among other things, this implies that if a (primal) *nonsmooth minimization* problem can be reformulated as a smooth convex-concave minimax optimization problem, then we can solve it in near-linear time instead of the best known super-linear time [4, 30]. In prior work, near-linear time DP algorithms with optimal rates for solving nonsmooth convex objectives only existed for generalized linear losses [9].

• The framework also enriches the cohort of near-linear time near-optimal private algorithms for smooth DP-SCO settings, which only contained SGD previously. Moreover, existing optimal DP algorithms for smooth DP-SCO rely on a stability result of SGD, which only holds for a restricted range of the smoothness parameter [24]. Such restrictions are avoided in our framework.

**Related Works:** Differentially private optimization has been an active research field for the past few years, and early works focused on the *empirical* problems [6, 46, 50, 44]. In addition to the output perturbation used in this paper, many existing works applied gradient perturbation to guarantee privacy. This method adds noise to each iteration of the algorithm and then applies moments accountant [1] or advanced composition [27] to analyze the overall privacy. Although gradient perturbation does not need the smoothness or convexity assumptions of the function and works for most iterative algorithms, it requires larger mini-batches [7] or longer training time [8] if one resorts to the privacy amplification via subsampling [10] to reduce the DP noise, resulting in super-linear gradient queries in existing methods [7, 49]. Here we mainly review previous results that achieve the optimal utility bound $\mathcal{O}(1/\sqrt{n} + \sqrt{d \log(1/\delta)}/(n\varepsilon))$ on the *population* loss—according to a lower-bound in Bassily et al. [6]—when solving a $d$ dimensional DP-SCO problem using $n$ samples with $(\varepsilon, \delta)$-DP guarantees.

In the smooth convex case, Bassily et al. [7] is the first to derive optimal rates for DP-SCO with complexity $\mathcal{O}(n^{3/2}\sqrt{\varepsilon})$ by gradient perturbation and stability of SGD [24]. Feldman et al. [22] provided two SGD-based linear-time algorithms: one uses privacy amplification by iteration [21] that only works for contractive updates; the second one uses phased output perturbation and stability of SGD. In the nonsmooth convex case, Bassily et al. [8] established the stability of SGD for nonsmooth functions and obtained a quadratic-time algorithm. Feldman et al. [22] leveraged the stability of ERM solutions [42] and achieved $\mathcal{O}(n^2 \log(1/\delta))$ complexity with phased output perturbation. Based on this phased framework, Asi et al. [4] and Kulkarni et al. [30] used gradient perturbation and improved the complexity to $\mathcal{O}(\min\{n^{3/2}, n^2/\sqrt{d}\})$ and $\mathcal{O}(\min\{n^{5/4}d^{1/8}, n^{3/2}/d^{1/8}\})$ respectively; Asi et al. [3] introduced a hypothetical linear-time algorithm assuming the existence of a low-biased estimator.

Moreover, Bassily et al. [9] gave a near-linear time algorithm for nonsmooth convex generalized linear losses using phased SGD [22] with smoothing techniques [37].

To the best of our knowledge, only few papers studied DP-SMO and all of them used gradient perturbation to guarantee privacy. Boob and Guzmán [11] analyzed stability of Extragradient [43] and proximal point methods [41] for smooth convex-concave functions, and their DP versions run with time $\mathcal{O}(n^2)$ and $\tilde{\mathcal{O}}(n^{3/2})$ respectively. Yang et al. [49] used the stability of SGDA [31] and obtained DP algorithms with complexity $\mathcal{O}(n^{3/2}\sqrt{\varepsilon})$ for the smooth convex-concave case and $\mathcal{O}(n^2)$ for the nonsmooth case, mirroring the guarantees of SGD for DP-SCO [7, 8]. Kang et al. [29] only focused on stability and generalization analysis of gradient perturbed SGDA and provided high-probability results. See Table 1 for a brief comparison.

The idea of using the stability of ERM for smooth convex optimization has also been exploited before. Attia and Koren [5] developed a black-box framework for smooth convex objectives that produces uniformly-stable algorithms while maintaining fast convergence rates. Lowy and Razaviyayn [33] considered DP-SCO by output perturbation but only provided near-linear time near-optimal algorithms for smooth strongly-convex losses. For smooth convex case, they achieved a sub-optimal rate $\mathcal{O}(1/\sqrt{n} + (\sqrt{d\log(1/\delta)}/(n\varepsilon))^{2/3})$ in near-linear time. In the setting of smooth strongly-convex–(strongly-)concave DP-SMO, they directly utilized its DP-SCO reformulation in the primal function to obtain the final guarantees. Therefore, their utility bound is simply on the primal risk, which is weaker than the primal-dual gap considered in this work. The algorithms run in near-linear time for the strongly-concave case and super-linear time $\mathcal{O}(n^{5/2})$ for the concave case. In contrast, we provide near-optimal near-linear time algorithms in all aforementioned settings.

**Notations:** We use $\|\cdot\|$ for the Euclidean norm of a vector and $|\cdot|$ for the absolute value or the cardinality of a set. A function $g : \mathbb{R}^d \to \mathbb{R}$ is $L$-Lipschitz if $|g(x_1) - g(x_2)| \leq L\|x_1 - x_2\|$ for $x_1, x_2$ in the domain of $g$. A function $h : \mathbb{R}^d \to \mathbb{R}$ is $\ell$-smooth if it is differentiable and $h(x_2) \leq h(x_1) + \nabla h(x_1)^\top (x_2 - x_1) + (\ell/2)\|x_1 - x_2\|^2$. A function $p : \mathbb{R}^d \to \mathbb{R}$ is convex if $p(\alpha x_1 + (1 - \alpha)x_2) \leq \alpha p(x_1) + (1 - \alpha)p(x_2)$ for all $\alpha \in [0, 1]$, and $p$ is $\mu$-strongly convex if $p(x) - (\mu/2)\|x\|^2$ is convex with $\mu > 0$. A function $q : \mathbb{R}^d \to \mathbb{R}$ is concave if $-q$ is convex and $\mu$-strongly concave if $-q$ is $\mu$-strongly convex. For a vector $x \in \mathbb{R}^d$, the notation $x + \mathcal{N}(0, \sigma^2 I_d)$ means $x + z$ for a random vector $z \sim \mathcal{N}(0, \sigma^2 I_d)$ sampled from the Gaussian distribution.

## 2 Preliminaries

We first provide some background on differential privacy and stochastic minimax optimization.

### 2.1 Differential Privacy

Differential Privacy (DP), introduced in Dwork et al. [18], measures privacy leakage of an algorithm.

**Definition 1.** For two datasets $S = \{\xi_i\}_{i=1}^n$ and $S' = \{\xi_i'\}_{i=1}^n$, we say the pair $(S, S')$ is *neighboring* if $\max\{|S \setminus S'|, |S' \setminus S|\} = 1$ and we denote neighboring datasets with $S \sim S'$. For an algorithm $\mathcal{A}$ and some privacy parameters $\varepsilon > 0$ and $\delta \in (0, 1)$, we say $\mathcal{A}$ satisfies $(\varepsilon, \delta)$-*differential privacy* if $\mathbb{P}(\mathcal{A}(S) \in A) \leq e^\varepsilon \mathbb{P}(\mathcal{A}(S') \in A) + \delta$ for all $S \sim S'$ and all subset $A$ in the range of $\mathcal{A}$.

In this work, we focus on the settings when $\varepsilon \in (0, 1)$ and $\delta \in (0, 1/n)$ given dataset of size $n$. Sensitivity is an important concept that makes designing $(\varepsilon, \delta)$-DP mechanisms straightforward.

**Definition 2.** Let $\mathcal{A}$ be some randomized algorithm operating on $S$ and outputting a vector in $\mathbb{R}^d$. If $\mathcal{A}$ has *sensitivity* $\Delta_{\mathcal{A}} := \sup_{S \sim S'} \|\mathcal{A}(S) - \mathcal{A}(S')\|$ with probability at least $1 - \delta$, then the *Gaussian mechanism* outputs $\mathcal{A}(S) + \mathcal{N}(0, (\Delta_{\mathcal{A}}\sqrt{2\log(1.25/\delta)}/\varepsilon)^2 I_d)$ and achieves $(\varepsilon, 2\delta)$-DP [17, 22].

The following basic composition rule of differential privacy will be used in the analysis.

**Lemma 2.1.** *If $\mathcal{A}_1$ is $(\varepsilon_1, \delta_1)$-DP and $\mathcal{A}_2$ is $(\varepsilon_2, \delta_2)$-DP, then $(\mathcal{A}_1, \mathcal{A}_2)$ is $(\varepsilon_1 + \varepsilon_2, \delta_1 + \delta_2)$-DP [17]. For a sequence of interactive algorithms $\{\mathcal{A}_k\}_{k=1}^K$ each satisfying $(\varepsilon_k, \delta_k)$-DP and operating on a subset $S_k$, if $S_k$'s are disjoint then the composition $(\mathcal{A}_1(S_1), \mathcal{A}_2(S_2), \ldots, \mathcal{A}_K(S_K))$ is $(\max_{k \in [K]} \varepsilon_k, \max_{k \in [K]} \delta_k)$-DP (known as parallel composition in McSherry [36]).*

---

**Algorithm 1** Output Perturbation for Strongly-Convex Minimization

---
**Input:** Dataset $S = \{\xi_i\}_{i=1}^n$, algorithm $\mathcal{A}$, DP parameters $(\varepsilon, \delta)$, strong-convexity parameter $\mu$.

1: Run the algorithm $\mathcal{A}$ on the smooth strongly-convex finite-sum problem $\min_{x \in \mathcal{X}} \hat{F}_S(x) = (1/n) \sum_{i=1}^n f(x; \xi_i)$ to obtain the output $\mathcal{A}(S)$ such that $\|\mathcal{A}(S) - \hat{x}_S^*\| \le L/(\mu n)$ with probability at least $1 - \delta/4$, where $\hat{x}_S^* = \arg\min_{x \in \mathcal{X}} \hat{F}_S(x)$ is the empirical optimal solution.

**Output:** $\tilde{x} = \mathcal{A}(S) + \mathcal{N}(0, \sigma^2 I_d)$ with $\sigma = 4L\sqrt{2\log(2.5/\delta)}/(\mu n \varepsilon)$.

---

## 2.2 Stochastic Minimax Optimization

The stochastic minimax (a.k.a. saddle point) optimization problem has the form:

$$\min_{x \in \mathcal{X}} \max_{y \in \mathcal{Y}} \ F(x, y) \triangleq \mathbb{E}_\xi[f(x, y; \xi)], \tag{1}$$

where $F$ is the population-level expectation of the stochastic continuous objective $f(\cdot, \cdot; \xi) : \mathbb{R}^{d_x} \times \mathbb{R}^{d_y} \to \mathbb{R}$ with closed convex domains $\mathcal{X} \subset \mathbb{R}^{d_x}$ and $\mathcal{Y} \subset \mathbb{R}^{d_y}$ whose stochasticity is captured by the random vector $\xi$. We are interested in the *population saddle point* $(x^*, y^*) \in \mathcal{X} \times \mathcal{Y}$ of the above problem such that $F(x^*, y) \le F(x^*, y^*) \le F(x, y^*)$ for all $(x, y) \in \mathcal{X} \times \mathcal{Y}$.

For some randomized algorithm with output $(\tilde{x}, \tilde{y})$, we measure its convergence rates by the *population strong duality gap* $\mathbb{E}[\max_{y \in \mathcal{Y}} F(\tilde{x}, y) - \min_{x \in \mathcal{X}} F(x, \tilde{y})]$ or the *population weak duality gap* $\max_{y \in \mathcal{Y}} \mathbb{E}[F(\tilde{x}, y)] - \min_{x \in \mathcal{X}} \mathbb{E}[F(x, \tilde{y})]$. Note that both duality gaps are always larger than 0, and a deterministic $(\tilde{x}, \tilde{y})$ is the saddle point if and only if the duality gaps are 0.

In practice, we usually do not have access to the distribution $P_\xi$ of the random vector $\xi$. Instead we are given a dataset $S = \{\xi_i\}_{i=1}^n$ with $n$ random vectors independently sampled from the distribution $P_\xi$. We define the *empirical* minimax optimization problem as

$$\min_{x \in \mathcal{X}} \max_{y \in \mathcal{Y}} \ \hat{F}_S(x, y) \triangleq \frac{1}{n} \sum_{i=1}^n f(x, y; \xi_i). \tag{2}$$

Similarly, we can define the *empirical saddle point* $(\hat{x}_S^*, \hat{y}_S^*)$ and the *empirical duality gap* w.r.t. the empirical function $\hat{F}_S$. Zhang et al. [51] established the stability and generalization properties of the empirical saddle point, which is essential to the design of our DP-SMO algorithms.

# 3 Differentially Private Stochastic Convex Optimization

As a warm-up, we start with a simpler problem of differentially private stochastic convex optimization (DP-SCO) to showcase our main ideas. We consider the following stochastic optimization:

$$\min_{x \in \mathcal{X}} \ F(x) \triangleq \mathbb{E}_\xi[f(x; \xi)],$$

where the stochastic function $f(\cdot, \xi) : \mathbb{R}^d \to \mathbb{R}$ is defined on a convex domain $\mathcal{X} \subset \mathbb{R}^d$ and $\xi$ is a random vector from an unknown distribution $P_\xi$. Given a dataset $S$ with $n$ i.i.d. samples from $P_\xi$, we develop a generic output perturbation framework with both privacy and population loss guarantees.

## 3.1 Near-Linear Time Algorithms for Smooth Strongly-Convex Functions

First we study the case when the objective function is strongly-convex with the following assumptions.

**Assumption 3.1.** *For any $\xi$, the function $f(x; \xi)$ is L-Lipschitz, $\ell$-smooth, and convex on the closed convex domain $\mathcal{X} \subset \mathbb{R}^d$.*

**Assumption 3.2.** *For any $\xi$, $f(x; \xi)$ satisfies Assumption 3.1 and it is $\mu$-strongly convex on $\mathcal{X}$.*

Shalev-Shwartz et al. [42] proved that the empirical optimal solution has bounded stability if the objective $f(\cdot; \xi)$ is strongly-convex and Lipschitz w.r.t. its domain (see Lemma A.2 in the appendix). As long as the output of some algorithm $\mathcal{A}$ is close enough to the empirical solution, we can show that $\mathcal{A}$ has bounded sensitivity, and thus standard Gaussian mechanism (Definition 2) can be applied to ensure differential privacy. This is formalized in Algorithm 1, where $\mathcal{A}$ is any algorithm for solving smooth strongly-convex finite-sum minimization problems. The theorem below shows that Algorithm 1 is $(\varepsilon, \delta)$-DP with near-optimal guarantees on the excess risk. A proof is provided in Appendix A.

---

**Algorithm 2** Phased Output Perturbation for Convex Minimization

---

**Input:** Dataset $S = \{\xi_i\}_{i=1}^n$, algorithm $\mathcal{A}$, DP parameters $(\varepsilon, \delta)$, regularizer $\mu$, initializer $x_0$.

1: Set $K = \log(n)$, $\bar{n} = n/K$ and $\tilde{x}_0 = x_0$.
2: **for** $k = 1, \cdots, K$ **do**
3:     Set $\mu_k = \mu \cdot 2^k$.
4:     Run the algorithm $\mathcal{A}$ on the smooth strongly-convex finite-sum minimization problem $\min_{x \in \mathcal{X}} \hat{F}_k(x) \triangleq (1/\bar{n}) \sum_{i=(k-1)\bar{n}+1}^{k\bar{n}} f(x; \xi_i) + (\mu_k/2)\|x - \tilde{x}_{k-1}\|^2$ to obtain the output $x_k$ such that $\|x_k - \hat{x}_k^*\| \leq L/(\mu_k \bar{n})$ with probability at least $1 - \delta/4$, where $\hat{x}_k^* = \arg\min_{x \in \mathcal{X}} \hat{F}_k(x)$ is the empirical optimal solution.
5:     $\tilde{x}_k = x_k + \mathcal{N}(0, \sigma_k^2 I_d)$ with $\sigma_k = 4L\sqrt{2\log(2.5/\delta)}/(\mu_k \bar{n}\varepsilon)$.
**Output:** $\tilde{x}_K$.

---

**Theorem 3.3.** *Under Assumption 3.2, Algorithm 1 is $(\varepsilon, \delta)$-DP and its output $\tilde{x}$ satisfies*

*(excess empirical risk)* $\qquad \mathbb{E}[\hat{F}_S(\tilde{x}) - \hat{F}_S(\hat{x}_S^*)] \leq 33L^2\kappa \cdot \dfrac{d\log(2.5/\delta)}{\mu n^2 \varepsilon^2},$

*(excess population risk)* $\qquad \mathbb{E}[F(\tilde{x}) - F(x^*)] \leq L^2\kappa\left(\dfrac{7}{\mu n} + \dfrac{48d\log(2.5/\delta)}{\mu n^2 \varepsilon^2}\right),$

*where $\kappa = \ell/\mu$ is the condition number and we assume $x^*$ and $\hat{x}_S^*$ are interior points of $\mathcal{X}$.*

**Remark 1.** Many algorithms for smooth strongly-convex finite-sum minimization problems guarantee $\mathbb{E}[\hat{F}_S(\mathcal{A}(S)) - \hat{F}_S(\hat{x}_S^*)] \leq \gamma$ with $\mathcal{O}(T(n, \kappa)\log(1/\gamma))$ gradient evaluations, where $n$ is the sample size and $\kappa$ is the condition number. Setting $1/\gamma = 32\mu n^2/(\delta^2 L^2)$ satisfies the requirements of $\mathcal{A}$ by Markov's inequality (see Appendix A). The gradient complexity of Algorithm 1 is $\mathcal{O}((n + \kappa)\log(n/\delta))$ for SVRG [26] or SARAH [38], and $\mathcal{O}((n + \sqrt{n\kappa})\log(n/\delta))$ for Katyusha [2].

As a comparison, most state-of-the-art private algorithms [44, 50, 14] for smooth strongly-convex functions only focus on empirical problems. The linear-time algorithms in Feldman et al. [22] achieve optimal population guarantees but only work for $\kappa \leq \tilde{\mathcal{O}}(n)$ since they utilize the stability of SGD. Instead, we provide a flexible framework that includes various base methods without the necessity to show their algorithm-specific stability. Further, near-linear time-complexity can be attained using fast variance reduction-based methods. Similar results existed in [33], but their base algorithm $\mathcal{A}$ only has in-expectation guarantees, which brings a critical challenge to the design of private mechanisms.

### 3.2 Near-Linear Time Algorithms for Smooth Convex Functions

Next, we study the convex setting. A direct method is to first reduce the convex problem to a strongly-convex one by adding a regularizer $(\mu/2)\|x\|^2$ to the objective and then apply Algorithm 1. However, this approach only achieves a sub-optimal rate [50, 33]. Inspired by the phased-ERM [22] method for nonsmooth convex losses, we show that a more sophisticated multi-phase Algorithm 2 using an increasing sequence of regularization parameters $\{\mu_k\}$ can achieve near-optimal guarantees on the population loss for smooth DP-SCO. The increasing regularization parameters ensure that both added DP noise and approximation errors coming from the regularizer can be properly controlled. The output of Algorithm 2 satisfies the following guarantees, as proved in Appendix A.

**Theorem 3.4.** *Let Assumption 3.1 hold. Suppose there exists at least one optimal solution $x^* \in \arg\min_{x \in \mathcal{X}} F(x)$ such that $\|x^*\| \leq D$. Algorithm 2 is $(\varepsilon, \delta)$-DP and its output $\tilde{x}_K$ satisfies*

$$\mathbb{E}[F(\tilde{x}_K) - F(x^*)] \leq 4LD \cdot \log(n)\left(\frac{1}{\sqrt{n}} + \frac{7\sqrt{d\log(2.5/\delta)}}{n\varepsilon}\right),$$

*for the excess population risk when setting $\mu = (L/D)\max\{1/\sqrt{n}, 14\log(n)\sqrt{d\log(2.5/\delta)}/(n\varepsilon)\}$.*

**Remark 2.** By Remark 1 and the fact that $\sum_{k=1}^K 1/\mu_k \leq \mathcal{O}((D/L)\sqrt{n})$, the total gradient complexity of Algorithm 2 is $\mathcal{O}((n + \sqrt{n}\ell D/L)\log(n/\delta))$ using SVRG [26] or SARAH [38], and $\mathcal{O}((n + n^{3/4}\sqrt{\ell D/L})\log(n/\delta))$ for Katyusha [2] (see Appendix A). We also point out that the smoothness assumption here is not necessary to derive the utility bound, but allows the use of fast accelerated algorithms for solving the regularized ERM problems.

---
**Algorithm 3** Output Perturbation for Strongly-Convex–Strongly-Concave Minimax Problems
---
**Input:** Dataset $S = \{\xi_i\}_{i=1}^n$, algorithm $\mathcal{A}$, DP parameters $(\varepsilon, \delta)$, SC parameters $(\mu_x, \mu_y)$.
  1: Run algorithm $\mathcal{A}$ on smooth strongly-convex–strongly-concave finite-sum saddle point problem $\min_{x \in \mathcal{X}} \max_{y \in \mathcal{Y}} \hat{F}_S(x, y) = (1/n) \sum_{i=1}^n f(x, y; \xi_i)$ to obtain the output $(\mathcal{A}_x(S), \mathcal{A}_y(S))$ such that with probability at least $1 - \delta/8$,

$$\mu_x \|\mathcal{A}_x(S) - \hat{x}_S^*\|^2 + \mu_y \|\mathcal{A}_y(S) - \hat{y}_S^*\|^2 \leq \frac{L^2}{\mu n^2},$$

    where $(\hat{x}_S^*, \hat{y}_S^*)$ is the saddle point of $\hat{F}_S(x, y)$ and we let $\mu := \min\{\mu_x, \mu_y\}$.
  2: Set $\sigma_x = (8L/(n\varepsilon))\sqrt{2 \log(5/\delta)/(\mu_x \mu)}$ and $\sigma_y = (8L/(n\varepsilon))\sqrt{2 \log(5/\delta)/(\mu_y \mu)}$.
**Output:** $\tilde{x} = \mathcal{A}_x(S) + \mathcal{N}(0, \sigma_x^2 \mathrm{I}_{d_x})$ and $\tilde{y} = \mathcal{A}_y(S) + \mathcal{N}(0, \sigma_y^2 \mathrm{I}_{d_y})$.
---

Bassily et al. [7] first proved optimal population guarantees for smooth convex functions, but their algorithm needs $\mathcal{O}(n^{3/2}\sqrt{\varepsilon})$ gradient queries. The algorithms in Feldman et al. [22] achieve linear time-complexity, and require that $\ell \leq \mathcal{O}((L/D) \max\{\sqrt{n}, \sqrt{d \log(1/\delta)}/\varepsilon\})$ rooted in the stability analysis of SGD. In contrast, our framework can achieve near-optimal population guarantees using any algorithm without additional restrictions to the smoothness parameter. By equipping with variance reduction-based methods, near-linear time-complexity can also be attained when $\ell \leq \mathcal{O}(\sqrt{n}L/D)$. To the best of our knowledge, this is the first time that sophisticated optimization algorithms besides SGD are proven to obtain near-optimal population guarantees in near-linear time. Additionally, in the regime that $\Omega(\sqrt{n}L/D) \leq \ell \leq \mathcal{O}(nL/D)$ where previous smooth DP-SCO algorithms [7, 22] fail to provide optimal guarantees, our framework still achieves a near-optimal rate with a better gradient complexity $\tilde{\mathcal{O}}(n^{3/4}\sqrt{\ell D/L})$ compared to the state-of-the-art nonsmooth DP-SCO algorithms [4, 30].

## 4 Differentially Private Stochastic Minimax Optimization

Using the same ideas as the minimization case in the previous section, we develop differentially private algorithms for stochastic minimax optimization in (1).

### 4.1 Near-Linear Time Algorithms for Smooth Strongly-Convex–Strongly-Concave Functions

First, we study the strongly-convex–strongly-concave (SC-SC) case with the following assumptions.

**Assumption 4.1.** *For any vector $\xi$, $f(x, y; \xi)$ is L-Lipschitz and $\ell$-smooth on the closed convex domain $\mathcal{X} \times \mathcal{Y} \subset \mathbb{R}^{d_x} \times \mathbb{R}^{d_y}$. Moreover, $f(\cdot, y; \xi)$ is convex on $\mathcal{X}$ for any $y \in \mathcal{Y}$, and $f(x, \cdot; \xi)$ is concave on $\mathcal{Y}$ for any $x \in \mathcal{X}$.*

**Assumption 4.2.** *For any vector $\xi$, $f(x, y; \xi)$ satisfies Assumption 4.1 and $f(\cdot, y; \xi)$ is $\mu_x$-strongly convex on $\mathcal{X}$ for any $y \in \mathcal{Y}$, and $f(x, \cdot; \xi)$ is $\mu_y$-strongly concave on $\mathcal{Y}$ for any $x \in \mathcal{X}$.*

In a manner similar to the minimization case, Zhang et al. [51] showed that the empirical saddle point is stable, which also implies its generalization error (see Lemma B.2 in the appendix). As a direct consequence, any algorithm $\mathcal{A}$ whose output is sufficiently close to the empirical saddle point has bounded sensitivity. This observation leads to Algorithm 3 for SC-SC DP-SMO with guarantees given in the theorem below. Here $\mathcal{A}$ can be any method for smooth SC-SC finite-sum minimax problems, and smoothness allows us to obtain efficient algorithms.

**Theorem 4.3.** *Under Assumption 4.2. Let saddle points $(\hat{x}_S^*, \hat{y}_S^*)$ and $(x^*, y^*)$ be interior points of the domain $\mathcal{X} \times \mathcal{Y}$. Then Algorithm 3 is $(\varepsilon, \delta)$-DP and its output $(\tilde{x}, \tilde{y})$ satisfies the following utility bounds on the empirical and population strong duality gap:*

$$\text{(empirical)} \qquad \mathbb{E}\left[\max_{y \in \mathcal{Y}} \hat{F}_S(\tilde{x}, y) - \min_{x \in \mathcal{X}} \hat{F}_S(x, \tilde{y})\right] \leq 257 L^2 (\kappa_x \kappa_y + \kappa) \cdot \frac{d \log(5/\delta)}{\mu n^2 \varepsilon^2},$$

$$\text{(population)} \qquad \mathbb{E}\left[\max_{y \in \mathcal{Y}} F(\tilde{x}, y) - \min_{x \in \mathcal{X}} F(x, \tilde{y})\right] \leq 3 L^2 (\kappa_x \kappa_y + \kappa)\left(\frac{3}{\mu n} + \frac{128 d \log(5/\delta)}{\mu n^2 \varepsilon^2}\right),$$

*where we let $\mu = \min\{\mu_x, \mu_y\}$, $\kappa_x = \ell/\mu_x$, $\kappa_y = \ell/\mu_y$, $\kappa = \ell/\mu$, and $d = \max\{d_x, d_y\}$.*

**Remark 3.** Existing methods for finite-sum saddle point problems output solutions such that $\mathbb{E}[\max_{y \in \mathcal{Y}} \hat{F}_S(\mathcal{A}_x(S), y) - \min_{x \in \mathcal{X}} \hat{F}_S(x, \mathcal{A}_y(S))] \le \gamma$ with complexity $\mathcal{O}(T(n, \kappa) \log(1/\gamma))$. Thus setting $1/\gamma = 16\mu n^2/(\delta L^2)$ satisfies the requirements of $\mathcal{A}$ (see Appendix B). The gradient complexity of Algorithm 3 is $\mathcal{O}(n\kappa \log(n/\delta))$ for Extragradient [43], $\mathcal{O}((n + \kappa^2) \log(n/\delta))$ for SVRG/SAGA [39], $\mathcal{O}((n + \sqrt{n}\kappa) \log(n/\delta))$ for Acc-SVRG/SAGA [39] and $\mathcal{O}((n + \sqrt{n\kappa_x\kappa_y} + n^{3/4}\sqrt{\kappa}) \log(n/\delta))$ for AL-SVRE [34] or Catalyst-Acc-SVRG [48].

The minimax problem is equivalent to a minimization problem on $x$ when the domain $\mathcal{Y}$ is restricted to a singleton. As a result, the lower-bound $\Omega(1/(\mu_x n) + d_x \log(1/\delta)/(\mu_x n^2 \varepsilon^2))$ of SC DP-SCO [6] trivially holds for SC-SC DP-SMO, and our results are near-optimal w.r.t. $n$ and $(\varepsilon, \delta)$. The flexible framework allows the use of off-the-shelf optimization algorithms for smooth SC-SC finite-sum minimax problems from a well-studied research community [48, 34] without being aware of the algorithm-specific stability bound. Based on this framework, we can produce the first near-linear time algorithms for smooth SC-SC DP-SMO with near-optimal guarantees on the duality gap. Similar rates were obtained in [33] on the population primal risk through the reduction to SC DP-SCO. However, the generalization error on the primal function does not always apply to the original minimax problem when the expectation and maximization cannot be exchanged.

## 4.2  Near-Linear Time Algorithms for Smooth Convex-Concave Functions

In this section, we focus on the more general case when the objective is smooth and convex-concave. We continue to build private algorithms using the stability and generalization results of empirical solutions. When the function is convex-concave, the following results for the regularized empirical problems can be applied by adding an SC-SC regularizer. Here, the stability bound is slightly tighter than the one in [51] and a proof is provided in Appendix B for completeness.

**Lemma 4.4.** *Consider a stochastic minimax problem such that $f(x, y; \xi)$ is convex-concave and L-Lipschitz with a $\mu_x$-strongly convex $\mu_y$-strongly concave regularizer $G(x, y)$. Let $\mu = \min\{\mu_x, \mu_y\}$ and denote the empirical saddle point of function $\hat{F}_S(x, y) + G(x, y)$ as $(\hat{x}_S^*, \hat{y}_S^*)$ given dataset $S$ with $n$ i.i.d. samples. Then for any neighboring datasets $S \sim S'$, we have*

$$\mu_x \|\hat{x}_S^* - \hat{x}_{S'}^*\|^2 + \mu_y \|\hat{y}_S^* - \hat{y}_{S'}^*\|^2 \le \frac{4L^2}{\mu n^2}.$$

*The stability result implies the generalization error of the empirical solution can be bounded as*

$$\max_{y \in \mathcal{Y}} \mathbb{E}\Big[F(\hat{x}_S^*, y) + G(\hat{x}_S^*, y)\Big] - \min_{x \in \mathcal{X}} \mathbb{E}\Big[F(x, \hat{y}_S^*) + G(x, \hat{y}_S^*)\Big] \le \frac{2\sqrt{2}L^2}{\mu n},$$

*measured by the population weak duality gap.*

Unlike DP-SCO, the interaction between the primal variable $x$ and the dual variable $y$ puts additional challenge in the analysis. In order to derive a near-optimal algorithm for convex-concave objectives, we need to carefully design the regularization parameters for the primal $x$ and the dual $y$ to control both approximation errors and DP noise. Based on the phased output perturbation framework, we propose a novel algorithm for convex-concave DP-SMO in Algorithm 4.

The small fixed parameter $\mu$ ensures the approximation error from adding regularization for the dual is properly bounded. The role of the increasing parameter $\mu_k$ is the same as Algorithm 2. We only add noise to the primal solutions and output $\tilde{x}_K$ in Algorithm 4. A natural way to derive a corresponding dual solution is to solve the smooth concave maximization problem $\max_{y \in \mathcal{Y}} \mathbb{E}[F(\tilde{x}_K, y)]$ with DP constraints. We instead provide an alternative method that is symmetric to Algorithm 4 by switching the role of the primal and the dual. This phased algorithm iteratively solves

$$\min_{x \in \mathcal{X}} \max_{y \in \mathcal{Y}} \left\{ \frac{1}{\bar{n}} \sum_{i=(k-1)\bar{n}+1}^{k\bar{n}} f(x, y; \xi_i) + \frac{\mu}{2} \|x\|^2 - \frac{\mu_k}{2} \|y - \tilde{y}_{k-1}\|^2 \right\},$$

at phase $k$ and only perturbs the dual variables. The detailed Algorithm 5 is provided in Appendix B. Its output $\tilde{y}_K$ together with the output $\tilde{x}_K$ of Algorithm 4 satisfy $(\varepsilon, \delta)$-DP by *basic composition* in Lemma 2.1. The guarantees are shown in the following theorem with detailed proofs in Appendix B.

---

**Algorithm 4** Phased Output Perturbation for Convex-Concave Minimax Problems

---

**Input:** Dataset $S = \{\xi_i\}_{i=1}^n$, algorithm $\mathcal{A}$, DP Parameters $(\varepsilon, \delta)$, regularizer $\mu$, initializer $x_0$.

1: Set $K = \log(n)$, $\bar{n} = n/K$ and $\tilde{x}_0 = x_0$.
2: **for** $k = 1, \cdots, K$ **do**
3:     Set $\mu_k = \mu \cdot 2^k$.
4:     Run the algorithm $\mathcal{A}$ on the smooth SC-SC finite-sum saddle point problem

$$\min_{x \in \mathcal{X}} \max_{y \in \mathcal{Y}} \ \hat{F}_k(x, y) \triangleq \frac{1}{\bar{n}} \sum_{i=(k-1)\bar{n}+1}^{k\bar{n}} f(x, y; \xi_i) + \frac{\mu_k}{2}\|x - \tilde{x}_{k-1}\|^2 - \frac{\mu}{2}\|y\|^2,$$

    to obtain the output $(x_k, y_k)$ such that with probability $1 - \delta/8$,

$$\mu_k\|x_k - \hat{x}_k^*\|^2 + \mu\|y_k - \hat{y}_k^*\|^2 \leq \frac{L^2}{\mu \bar{n}^2},$$

    where $(\hat{x}_k^*, \hat{y}_k^*)$ is the saddle point of the regularized empirical function $\hat{F}_k(x, y)$.

5:     $\tilde{x}_k = x_k + \mathcal{N}(0, \sigma_k^2 I_{d_x})$ with $\sigma_k = (8L/(\bar{n}\varepsilon))\sqrt{2\log(5/\delta)/(\mu_k\mu)}$.

**Output:** $\tilde{x}_K$.

---

**Theorem 4.5.** *Let Assumption 4.1 hold. Suppose $\max\{\|x\|, \|y\|\} \leq D$ for all $x \in \mathcal{X}$ and $y \in \mathcal{Y}$. Then the composition of Algorithm 4 and 5 is $(\varepsilon, \delta)$-DP and the output $(\tilde{x}_K, \tilde{y}_K)$ satisfies the following bound on the population weak duality gap:*

$$\max_{y \in \mathcal{Y}} \mathbb{E}[F(\tilde{x}_K, y)] - \min_{x \in \mathcal{X}} \mathbb{E}[F(x, \tilde{y}_K)] \leq 16LD \cdot \log^2(n)\left(\frac{1}{\sqrt{n}} + \frac{5\sqrt{d\log(5/\delta)}}{n\varepsilon}\right),$$

*when setting $\mu = (L/D)\max\left\{2/\sqrt{n}, 13\log(n)\sqrt{d\log(5/\delta)}/(n\varepsilon)\right\}$, where $d = \max\{d_x, d_y\}$.*

**Remark 4.** By Remark 3, we can set $1/\gamma = 16\mu\bar{n}^2/(\delta L^2)$ to satisfy the requirements of $\mathcal{A}$ in each phase. For example, if we use Extragradient [43], the total complexity is $2\sum_{k=1}^K \mathcal{O}((\bar{n}(\ell + \mu_k)/\mu)\log(\bar{n}/\delta)) = \mathcal{O}((n^{3/2}\ell D/L + n^2)\log(n/\delta))$. After similar calculations, the complexity becomes $\mathcal{O}((n\ell D/L + n^{5/4})\log(n/\delta))$ if $\mathcal{A}$ is AL-SVRE [34] or Catalyst-Acc-SVRG [48] noticing that the condition number $(\ell + \mu_k)/\mu$ can be as large as $\mathcal{O}(n)$.

**Remark 5.** To achieve near-linear time-complexity, we utilize the special structures of our regularized problems in Algorithm 4 and 5. (i) Objective functions of the regularized problems are separable with different smoothness parameters w.r.t. $x$ and $y$. Applying the algorithm in Jin et al. [25] that is optimized for such special case, we can achieve the overall complexity $\mathcal{O}((n + n\ell D/L)\log(n/\delta))$. (ii) The regularization terms with potentially large smoothness parameters are prox-friendly [2]. SVRG/SAGA and its accelerated version [39] which leverage this property can be used, and the complexity only depends on the smoothness parameter of $f(x, y; \xi)$. The total complexity is $\mathcal{O}((n + n(\ell D/L)^2)\log(n/\delta))$ for SVRG/SAGA, and $\mathcal{O}((n + n\ell D/L)\log(n/\delta))$ for Acc-SVRG/SAGA, noticing that now the condition number $\ell/\mu$ is only $\mathcal{O}(\sqrt{n}\ell D/L)$. More details are provided in Appendix B.

Algorithm 4 can be extended to the smooth C-SC case or the SC-C case, where near-linear time near-optimal algorithms are also provided (see Appendix B). According to the lower-bound discussed in [11], our utility guarantee is optimal up to logarithmic terms. Although the primal problem $\min_{x \in \mathcal{X}} \Phi(x) \triangleq \max_{y \in \mathcal{Y}} F(x, y)$ is not necessarily smooth when $f(x, y; \xi)$ is smooth and convex-concave, we provide several instances of the flexible framework that achieve near-optimal rates in near-linear time. This improves upon current results for DP-SMO [11, 49] and gives a new example where near-linear time algorithms are available for nonsmooth DP-SCO. Our results suggest that for nonsmooth DP-SCO problems, if there exists some smooth convex-concave minimax reformulation, then near-optimal rates can be attained in near-linear time. For example, the nonsmooth convex problem $\min_{x \in \mathbb{R}^{d_x}} \|Ax - b\|_1$ given $A \in \mathbb{R}^{d_y \times d_x}$ and $b \in \mathbb{R}^{d_y}$ is equivalent to the smooth convex-concave minimax problem $\min_{x \in \mathbb{R}^{d_x}} \max_{y \in \mathbb{R}^{d_y}, \|y\|_\infty \leq 1} y^\top (Ax - b)$. However, it remains an open question whether optimal utility bound can be obtained in linear time for general nonsmooth problems.

---

[2]This means the proximal operator is easy to compute. See Palaniappan and Bach [39] for more details.

## 5 Conclusion

We provide a general framework for both smooth DP-SCO and DP-SMO problems with the near-optimal privacy-utility trade-off. The flexible framework allows to bring various off-the-shelf, fast convergent non-private optimization algorithms into the DP domain. Using the framework, we enrich the class of near-linear time algorithms for smooth DP-SCO and provide the first near-linear time algorithms for smooth DP-SMO. For future work, it is interesting to study whether the logarithmic terms in gradient complexity and the final utility bound of our algorithms can be further removed. Another direction is to derive simpler private algorithms for smooth convex-concave DP-SMO based on the special structure of the regularized problems. Moreover, we believe our framework also opens the door to simpler derivatives with possibly improved complexities for nonsmooth DP-SCO, nonsmooth DP-SMO, and even more general forms of optimization problems such as variational inequalities and games. We will leave these for future investigation.

## Acknowledgments and Disclosure of Funding

We would like to thank anonymous reviewers for the insightful feedback. L.Z. gratefully acknowledges funding by the Max Planck ETH Center for Learning Systems (CLS). This work does not relate to the current position of K.T. at Amazon. S.O. is supported in part by NSF grants CNS-2002664, IIS-1929955, and CCF-2019844 as a part of NSF Institute for Foundations of Machine Learning (IFML). N.H. is supported by ETH research grant funded through ETH Zurich Foundations and NCCR Automation funded through Swiss National Science Foundations; part of the work was done while N.H. was visiting the Simons Institute for the Theory of Computing.

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
