# A   Differentially Private Stochastic Convex Optimization

In this section, we provide analyses of our near-linear time algorithms for DP-SCO with near-optimal utility guarantees on the excess (population) risk. We first present some helpful lemmas that already exist in the literature and give their proofs for completeness.

## A.1   Supporting Lemmas

In the phased algorithms for both convex minimization and convex-concave minimax problems, we *interactively* [17] access a dataset multiple times where a future output is allowed to depend on all the past outputs. However, since each phase only accesses a disjoint partition of the dataset, we can use the parallel composition in Lemma 2.1 to guarantee differential privacy. Here we provide a specific form of it that can be directly applied to our algorithms.

**Lemma A.1.** *(Parallel Composition [36]) Given a dataset $S$ and its disjoint partition $S = \bigcup_{k=1}^{K} S_k$, define the mechanisms as $\mathcal{A}_1 = \mathcal{A}_1(S_1), \mathcal{A}_2 = \mathcal{A}_2(S_2; \mathcal{A}_1), \cdots, \mathcal{A}_K = \mathcal{A}_K(S_K; \mathcal{A}_{K-1})$. Suppose each mechanism $\mathcal{A}_k(S_k; \mathcal{A}_{k-1})$ is $(\varepsilon, \delta)$-DP w.r.t. the set $S_k$ for $k = 1, \cdots, K$, then the composition $\mathcal{A}_K$ is $(\varepsilon, \delta)$-DP w.r.t. the full dataset $S$.*

*Proof.* For neighboring datasets $S \sim S'$, without loss of generality we let $S = \{\xi_1, \cdots, \xi_i, \cdots, \xi_n\}$ and $S' = \{\xi_1, \cdots, \xi_i', \cdots, \xi_n\}$, where $\xi_i$ and $\xi_i'$ are sampled independently. That is, the only difference between the datasets comes from $\xi_i$ and $\xi_i'$, and the remaining samples are the same. As a result, for the disjoint partitions $S = \bigcup_{k=1}^{K} S_k$ and $S' = \bigcup_{k=1}^{K} S_k'$, we can conclude that there is only one pair $S_j \sim S_j'$ for some $j \in \{1, \cdots, K\}$, and that $S_k = S_k'$ for $k \neq j$. Then we have that

$$
\mathbb{P}(\mathcal{A}_K(S)) = \prod_{k=1}^{K} \mathbb{P}(\mathcal{A}_k(S_k)|\mathcal{A}_{k-1})
$$

$$
= \mathbb{P}(\mathcal{A}_j(S_j)|\mathcal{A}_{j-1}) \prod_{k=1, k \neq j}^{K} \mathbb{P}(\mathcal{A}_k(S_k')|\mathcal{A}_{k-1})
$$

$$
\leq \left( e^\varepsilon \mathbb{P}(\mathcal{A}_j(S_j')|\mathcal{A}_{j-1}) + \delta \right) \prod_{k=1, k \neq j}^{K} \mathbb{P}(\mathcal{A}_k(S_k')|\mathcal{A}_{k-1})
$$

$$
\leq e^\varepsilon \mathbb{P}(\mathcal{A}_K(S')) + \delta.
$$

By Definition 1 of $(\varepsilon, \delta)$-differential privacy, the proof is complete. $\square$

Next, we restate the stability and generalization results of the empirical risk minimization shown in Shalev-Shwartz et al. [42]. Lemma A.2 considers the case when the objective $f(x; \xi)$ is strongly-convex and Lemma A.3 studies the case when $f(x; \xi)$ is convex with a strongly-convex regularizer. Both results depend on the Lipschitzness parameter of the objectives, making $L$ a critical parameter in private algorithms [7, 22, 49]. In practice, any estimate of the upper bound of Lipschitz constant $L$ can be used, e.g., see methods in [45, 20].

Here we only give a detailed proof of the regularized version. The proof of Lemma A.2 can be derived similarly and we omit it. It is worth mentioning that Lemma A.3 does not require the Lipschitzness of the regularizer, and is not a trivial extension of its unregularized version.

**Lemma A.2.** *[42, Theorem 6] Consider a stochastic optimization problem such that $f(x; \xi)$ is $\mu$-strongly convex and $L$-Lipschitz w.r.t. $x \in \mathcal{X}$ for any $\xi$. Given a dataset $S$ with $n$ i.i.d. samples, denote the empirical optimal solution as $\hat{x}_S^* = \arg\min_{x \in \mathcal{X}} \hat{F}_S(x) \triangleq (1/n) \sum_{i=1}^{n} f(x; \xi_i)$. Then for any neighboring datasets $S \sim S'$, we have that*

$$
\|\hat{x}_S^* - \hat{x}_{S'}^*\| \leq \frac{2L}{\mu n}.
$$

*The stability result also implies the generalization error of $\hat{x}_S^*$ can be bounded as*

$$
\mathbb{E}[F(\hat{x}_S^*) - F(x^*)] \leq \frac{2L^2}{\mu n},
$$

*where $x^* = \arg\min_{x \in \mathcal{X}} F(x) \triangleq \mathbb{E}_\xi[f(x; \xi)]$ is the population optimal solution.*

**Lemma A.3.** *[42, Theorem 7] Under the same settings as Lemma A.2. Consider the case when $f(x;\xi)$ is convex and $L$-Lipschitz with a $\mu$-strongly convex regularizer $G(x)$. Denote the empirical optimal solution as $\hat{x}_S^* = \arg\min_{x \in \mathcal{X}}\{\hat{F}_S(x) + G(x)\}$. Then for any neighboring datasets $S \sim S'$,*

$$\|\hat{x}_S^* - \hat{x}_{S'}^*\| \le \frac{2L}{\mu n}.$$

*The stability result also implies the generalization error of the empirical solution:*

$$\mathbb{E}\left[F(\hat{x}_S^*) + G(\hat{x}_S^*)\right] - \mathbb{E}\left[\min_{x \in \mathcal{X}}\{F(x) + G(x)\}\right] \le \frac{2L^2}{\mu n},$$

*measured by the excess population risk.*

*Proof.* Without loss of generality, we let neighboring datasets $S = \{\xi_1, \cdots, \xi_i, \cdots, \xi_n\}$ and $S_i' = \{\xi_1, \cdots, \xi_i', \cdots, \xi_n\}$, where $\xi_i$ and $\xi_i'$ are sampled independently.

Since $\hat{F}_S(x) + G(x)$ is $\mu$-strongly convex and $\hat{x}_S^*$ is the optimal solution, we have that

$$\hat{F}_S(\hat{x}_{S_i'}^*) + G(\hat{x}_{S_i'}^*) \ge \hat{F}_S(\hat{x}_S^*) + G(\hat{x}_S^*) + \frac{\mu}{2}\|\hat{x}_S^* - \hat{x}_{S_i'}^*\|^2.$$

Similarly, by strong-convexity of $\hat{F}_{S_i'} + G(x)$ and optimality of $\hat{x}_{S_i'}^*$, we have that

$$\hat{F}_{S_i'}(\hat{x}_S^*) + G(\hat{x}_S^*) \ge \hat{F}_{S_i'}(\hat{x}_{S_i'}^*) + G(\hat{x}_{S_i'}^*) + \frac{\mu}{2}\|\hat{x}_S^* - \hat{x}_{S_i'}^*\|^2.$$

Summing up the above two equations, we can obtain

$$
\begin{aligned}
\mu\|\hat{x}_S^* - \hat{x}_{S_i'}^*\|^2 &\le \hat{F}_S(\hat{x}_{S_i'}^*) - \hat{F}_{S_i'}(\hat{x}_{S_i'}^*) + \hat{F}_{S_i'}(\hat{x}_S^*) - \hat{F}_S(\hat{x}_S^*) \\
&= \frac{1}{n}[f(\hat{x}_{S_i'}^*;\xi_i) - f(\hat{x}_{S_i'}^*;\xi_i')] + \frac{1}{n}[f(\hat{x}_S^*;\xi_i') - f(\hat{x}_S^*;\xi_i)] \\
&= \frac{1}{n}[f(\hat{x}_{S_i'}^*;\xi_i) - f(\hat{x}_S^*;\xi_i)] + \frac{1}{n}[f(\hat{x}_S^*;\xi_i') - f(\hat{x}_{S_i'}^*;\xi_i')] \\
&\le \frac{2L}{n}\|\hat{x}_S^* - \hat{x}_{S'}^*\|,
\end{aligned}
\tag{3}
$$

where the first equality holds since for any $x$,

$$
\begin{aligned}
\hat{F}_S(x) - \hat{F}_{S_i'}(x) &= \frac{1}{n}\sum_{j=1}^n f(x;\xi_j) - \frac{1}{n}\left(\sum_{j=1, j \ne i}^n f(x;\xi_j) + f(x;\xi_i')\right) \\
&= \frac{1}{n}[f(x;\xi_i) - f(x;\xi_i')],
\end{aligned}
\tag{4}
$$

and the last inequality follows from $L$-Lipschitzness of $f(x;\xi)$. As a result of (3), we obtain the stability of empirical solutions as

$$\|\hat{x}_S^* - \hat{x}_{S_i'}^*\| \le \frac{2L}{\mu n}.\tag{5}$$

For the generalization error, we follow the standard results on stability and generalization. Let $x^* = \arg\min_{x \in \mathcal{X}}\{F(x) + G(x)\}$ for notation simplicity, and then

$$
\begin{aligned}
\mathbb{E}[F(\hat{x}_S^*) + G(\hat{x}_S^*)] - \mathbb{E}[F(x^*) + G(x^*)] &\overset{(a)}{=} \mathbb{E}[F(\hat{x}_S^*) + G(\hat{x}_S^*)] - \mathbb{E}[\hat{F}_S(x^*) + G(x^*)] \\
&\overset{(b)}{\le} \mathbb{E}[F(\hat{x}_S^*) + G(\hat{x}_S^*)] - \mathbb{E}[\hat{F}_S(\hat{x}_S^*) + G(\hat{x}_S^*)] \\
&\overset{(c)}{=} \mathbb{E}\left[\frac{1}{n}\sum_{i=1}^n F(\hat{x}_{S_i'}^*) - \frac{1}{n}\sum_{i=1}^n f(\hat{x}_S^*;\xi_i)\right] \\
&\overset{(d)}{=} \frac{1}{n}\sum_{i=1}^n \mathbb{E}[f(\hat{x}_{S_i'}^*;\xi_i) - f(\hat{x}_S^*;\xi_i)] \\
&\overset{(e)}{\le} \frac{2L^2}{\mu n},
\end{aligned}
$$

where $(a)$ holds since $x^*$ is independent of $S$, $(b)$ follows by the optimality of $\hat{x}_S^*$, $(c)$ uses the fact that $\hat{x}_S^*$ and $\hat{x}_{S_i'}^*$ have the same distribution for each $i$, $(d)$ is true because $S_i'$ is independent of $\xi_i$ and $(e)$ uses Lipschitzness of $f(x;\xi)$ and stability bound (5). $\qquad\square$

## A.2 Near-Linear Time Algorithms for Smooth Strongly-Convex Functions

For strongly-convex functions, Algorithm 1 achieves near-optimal excess risk bounds with near-linear time-complexity. The proof of Theorem 3.3 that gives its guarantees is provided below.

*Proof of Theorem 3.3.* We first prove the privacy guarantee. Given neighboring datasets $S \sim S'$, the sensitivity of $\mathcal{A}$ in Algorithm 1 is bounded as

$$\|\mathcal{A}(S) - \mathcal{A}(S')\| \leq \|\mathcal{A}(S) - \hat{x}_S^*\| + \|\hat{x}_S^* - \hat{x}_{S'}^*\| + \|\hat{x}_{S'}^* - \mathcal{A}(S')\|$$
$$\leq \frac{4L}{\mu n}, \tag{6}$$

with probability $1 - \delta/2$ by the union bound, where the last inequality follows from the stability of empirical solutions in Lemma A.2 and guarantees of algorithm $\mathcal{A}$. Then by Gaussian mechanism in Definition 2, Algorithm 1 is $(\varepsilon, \delta)$-DP when setting $\sigma = 4L\sqrt{2\log(2.5/\delta)}/(\mu n \varepsilon)$.

We then give the guarantees for the output $\tilde{x}$. Since $\hat{F}_S$ is $\ell$-smooth and $\hat{x}_S^*$ is in the interior of $\mathcal{X}$, the excess empirical risk satisfies

$$\mathbb{E}[\hat{F}_S(\tilde{x}) - \hat{F}_S(\hat{x}_S^*)] \leq \frac{\ell}{2}\mathbb{E}\|\tilde{x} - \hat{x}_S^*\|^2$$
$$\leq \ell\left(\mathbb{E}\|\tilde{x} - \mathcal{A}(S)\|^2 + \mathbb{E}\|\mathcal{A}(S) - \hat{x}_S^*\|^2\right)$$
$$\leq \ell\left(\frac{32L^2 \cdot d\log(2.5/\delta)}{\mu^2 n^2 \varepsilon^2} + \frac{\delta^2 L^2}{16\mu^2 n^2}\right)$$
$$< 33L^2\kappa \cdot \frac{d\log(2.5/\delta)}{\mu n^2 \varepsilon^2}, \tag{7}$$

where the third inequality is due to the guarantee of $\mathcal{A}$ in Remark 1 and the choice of $\sigma$, and the last inequality follows from the standard settings that $\varepsilon < 1$, $d \geq 1$ and $\delta < 1/n$, and $\kappa = \ell/\mu$ is the condition number. Similarly for the excess population risk, we have that

$$\mathbb{E}[F(\tilde{x}) - F(x^*)] \leq \frac{\ell}{2}\mathbb{E}\|\tilde{x} - x^*\|^2$$
$$\leq \frac{3\ell}{2}\left(\mathbb{E}\|\tilde{x} - \mathcal{A}(S)\|^2 + \mathbb{E}\|\mathcal{A}(S) - \hat{x}_S^*\|^2 + \mathbb{E}\|\hat{x}_S^* - x^*\|^2\right)$$
$$\leq \ell\left(\frac{48L^2 \cdot d\log(2.5/\delta)}{\mu^2 n^2 \varepsilon^2} + \frac{3\delta^2 L^2}{32\mu^2 n^2} + \frac{6L^2}{\mu^2 n}\right)$$
$$< L^2\kappa\left(\frac{7}{\mu n} + \frac{48d\log(2.5/\delta)}{\mu n^2 \varepsilon^2}\right),$$

where we use the fact that $(\mu/2)\mathbb{E}\|\hat{x}_S^* - x^*\|^2 \leq \mathbb{E}[F(\hat{x}_S^*) - F(x^*)]$ by strong-convexity and the generalization bound in Lemma A.2. Note that we can still obtain the bound for $\mathbb{E}\|\tilde{x} - \hat{x}_S^*\|^2$ and $\mathbb{E}\|\tilde{x} - x^*\|^2$ if we do not assume $\hat{x}_S^*$ and $x^*$ are interior points. $\square$

**Near-optimality:** The utility bounds on the excess risk have an extra $\kappa$ dependence compared to the optimal bound [6, 7]. When the problem is not ill-conditioned, we can achieve the optimal rate.

**Gradient Complexity:** Algorithm 1 requires high-probability convergence of $\mathcal{A}$ such that $\|\mathcal{A}(S) - \hat{x}_S^*\| \leq L/(\mu n)$ with probability $1 - \delta/4$. Existing algorithms for smooth strongly-convex finite-sum problems guarantee that $\mathbb{E}[\hat{F}_S(\mathcal{A}(S)) - \hat{F}_S(\hat{x}_S^*)] \leq \gamma$ with $\mathcal{O}(T(n, \kappa)\log(1/\gamma))$ gradient evaluations. For example, SVRG [26] and SARAH [38] have gradient complexity $\mathcal{O}((n + \kappa)\log(1/\gamma))$ and Katyusha [2] needs $\mathcal{O}((n + \sqrt{n\kappa})\log(1/\gamma))$ gradient queries, where $\kappa = \ell/\mu$ is the condition number. Since $\mathbb{E}\|\mathcal{A}(S) - \hat{x}_S^*\| \leq \sqrt{2\gamma/\mu}$ by strong-convexity of $\hat{F}_S(x)$ and Jensen's inequality, we can then apply Markov's inequality for $\|\mathcal{A}(S) - \hat{x}_S^*\| \geq 0$ and obtain that

$$\mathbb{P}\left(\|\mathcal{A}(S) - \hat{x}_S^*\| \geq \frac{L}{\mu n}\right) \leq \frac{n\sqrt{2\gamma\mu}}{L}.$$

Setting $\gamma = \delta^2 L^2/(32\mu n^2)$, the RHS becomes $\delta/4$ and the requirement of $\mathcal{A}$ is satisfied. This implies the complexity of Algorithm 1 is $\mathcal{O}(T(n, \kappa)\log(n/\delta))$ as discussed in Remark 1. Therefore, we achieve the linear-time gradient complexity up to logarithmic factors in $n/\delta$.

## A.3 Near-Linear Time Algorithms for Smooth Convex Functions

This section contains the proof of Theorem 3.4 that gives analyses of the near-optimal Algorithm 2 for convex functions. The core is the following lemma that leverages the generalization properties of regularized empirical problems.

**Lemma A.4.** *For $k = 1, \cdots, K$, by the settings and notations in Algorithm 2, we have that*

$$\mathbb{E}[F(\hat{x}_k^*) - F(\hat{x}_{k-1}^*)] \leq \frac{\mu_k}{2} \mathbb{E} \|\hat{x}_{k-1}^* - \tilde{x}_{k-1}\|^2 + \frac{2L^2}{\mu_k \bar{n}},$$

*where $\hat{x}_k^*$ is the optimal solution of the regularized empirical function $\hat{F}_k(x)$ and $\hat{x}_0^*$ is defined later in the proof of Theorem 3.4.*

*Proof.* Applying the generalization results in Lemma A.3 for $\hat{F}_k$ with regularization term $(\mu_k/2)\|x - \tilde{x}_{k-1}\|^2$ and dataset $S_k := \{\xi_i\}_{i=(k-1)\bar{n}+1}^{k\bar{n}}$, we have that for any $x \in \mathcal{X}$,

$$\mathbb{E}\Big[F(\hat{x}_k^*) + \frac{\mu_k}{2}\|\hat{x}_k^* - \tilde{x}_{k-1}\|^2\Big] \leq \mathbb{E}\Big[\min_{x' \in \mathcal{X}} \Big\{F(x') + \frac{\mu_k}{2}\|x' - \tilde{x}_{k-1}\|^2\Big\}\Big] + \frac{2L^2}{\mu_k \bar{n}}$$

$$\leq \mathbb{E}\Big[F(x) + \frac{\mu_k}{2}\|x - \tilde{x}_{k-1}\|^2\Big] + \frac{2L^2}{\mu_k \bar{n}}.$$

Setting $x = \hat{x}_{k-1}^*$, the proof is done since $\|\hat{x}_k^* - \tilde{x}_{k-1}\|^2 \geq 0$. $\qquad\square$

Now we are ready to prove Theorem 3.4.

*Proof of Theorem 3.4.* We first give the guarantees of each phase in Algorithm 2 by a similar proof as Theorem 3.3 in the previous section (see (6) and (7)). For phase $1 \leq k \leq K$, by the stability of $\hat{x}_k^*$ in Lemma A.3 and the requirement of output $x_k$ in Algorithm 2, we know $x_k$ has sensitivity bounded by $4L/(\mu_k \bar{n})$ with probability at least $1 - \delta/2$, and then setting $\sigma_k = 4L\sqrt{2\log(2.5/\delta)}/(\mu_k \bar{n}\varepsilon)$ guarantees $(\varepsilon, \delta)$-DP. As a result, we can obtain that

$$\mathbb{E}\|\tilde{x}_k - \hat{x}_k^*\|^2 \leq 2\mathbb{E}\|\tilde{x}_k - x_k\|^2 + 2\mathbb{E}\|x_k - \hat{x}_k^*\|^2$$

$$\leq 2d\sigma_k^2 + \frac{\delta^2 L^2}{8\mu_k^2 \bar{n}^2}$$

$$< 65L^2 \cdot \frac{d\log(2.5/\delta)}{\mu_k^2 \bar{n}^2 \varepsilon^2}. \tag{8}$$

Then we analyze the full algorithm. By the parallel composition in Lemma A.1, Algorithm 2 is $(\varepsilon, \delta)$-DP since we use disjoint datasets for different phases and each phase is $(\varepsilon, \delta)$-DP. For the excess population risk of the output $\tilde{x}_K$, we decompose the error as

$$\mathbb{E}[F(\tilde{x}_K) - F(x^*)] = \mathbb{E}[F(\tilde{x}_K) - F(\hat{x}_K^*)] + \sum_{k=1}^{K} \mathbb{E}[F(\hat{x}_k^*) - F(\hat{x}_{k-1}^*)], \tag{9}$$

where $x^* \in \arg\min_{x \in \mathcal{X}} F(x)$ is the population optimal solution, $\hat{x}_k^*$ is the optimal solution of the regularized empirical function $\hat{F}_k(x)$ in Algorithm 2, and we let $\hat{x}_0^* = x^*$ only for simplicity of the analysis. For the first term in the RHS of (9), we have that

$$\mathbb{E}[F(\tilde{x}_K) - F(\hat{x}_K^*)] \overset{(a)}{\leq} L\sqrt{\mathbb{E}\|\tilde{x}_K - \hat{x}_K^*\|^2}$$

$$\overset{(b)}{<} 9L^2 \cdot \frac{\sqrt{d\log(2.5/\delta)}}{\mu_K \bar{n}\varepsilon}$$

$$\overset{(c)}{\leq} 9LD \cdot \frac{\sqrt{d\log(2.5/\delta)}}{\bar{n}\varepsilon},$$

where $(a)$ holds by $L$-Lipschitzness of $F(x)$ and Cauchy–Schwarz inequality, $(b)$ uses (8) and $(c)$ follows from the settings that $\mu_K = \mu n$ and $\mu \geq L/(D\sqrt{n})$. Therefore, with Lemma A.4 to handle the second term in the RHS of (9), we obtain that

$$\mathbb{E}[F(\tilde{x}_K) - F(x^*)] \leq 9LD \cdot \frac{\sqrt{d\log(2.5/\delta)}}{\bar{n}\varepsilon} + \sum_{k=1}^{K}\left(\frac{\mu_k}{2}\mathbb{E}\|\hat{x}_{k-1}^* - \tilde{x}_{k-1}\|^2 + \frac{2L^2}{\mu_k\bar{n}}\right)$$

$$\leq 9LD \cdot \frac{\sqrt{d\log(2.5/\delta)}}{\bar{n}\varepsilon} + \sum_{k=2}^{K}\frac{65\mu_k}{2\mu_{k-1}}\frac{L^2 d\log(2.5/\delta)}{\mu_{k-1}\bar{n}^2\varepsilon^2} + \sum_{k=1}^{K}\frac{2L^2}{\mu_k\bar{n}} + \frac{\mu_1}{2}\|x^* - x_0\|^2$$

$$\leq 9LD \cdot \frac{\sqrt{d\log(2.5/\delta)}}{\bar{n}\varepsilon} + \frac{65L^2 d\log(2.5/\delta)}{\mu\bar{n}^2\varepsilon^2} + \frac{2L^2}{\mu\bar{n}} + \mu\|x^* - x_0\|^2$$

$$\leq 4LD \cdot \log(n)\left(\frac{1}{\sqrt{n}} + \frac{7\sqrt{d\log(2.5/\delta)}}{n\varepsilon}\right),$$

where the second inequality uses the guarantees of $\tilde{x}_{k-1}$ in (8) for $k \geq 2$ and the settings that $\hat{x}_0^* = x^*, \tilde{x}_0 = x_0$, the third inequality follows from the choice that $\mu_k = \mu \cdot 2^k$ and the last inequality holds since $\mu = (L/D)\max\{1/\sqrt{n}, 14\log(n)\sqrt{d\log(2.5/\delta)}/(n\varepsilon)\}$ and $\|x^*\|^2 \leq D^2$ when the initialization is $x_0 = 0$. $\qquad\square$

**Near-optimality:** The utility bound on the excess population risk has an extra logarithmic term in $n$, which can be removed by a different parameter choice. For example in phased-ERM [22], the regularization parameter increases as $\mu_k = \mu \cdot 2^{3k}$ across different phases, and the size of partitioned datasets decreases as $n_k = n/2^k$. However, the decreasing data size may not give us near-optimal algorithms for convex-concave minimax problems. To be consistent, we also use a fixed data size for the convex minimization case, despite this additional logarithmic factor compared to phased-ERM.

**Gradient Complexity:** Remark 1 suggests that the complexity of each phase is $\mathcal{O}(T(\bar{n}, (\ell + \mu_k)/\mu_k)\log(1/\gamma_k))$ with $\gamma_k = \delta^2 L^2/(32\mu_k\bar{n}^2)$ when solving a $\mu_k$-strongly convex, $(\ell+\mu_k)$-smooth finite-sum problems with sample size $\bar{n}$. Then the total complexity is just $\sum_{k=1}^{K}\mathcal{O}(T(\bar{n}, \ell/\mu_k + 1)\log(n/\delta))$. Therefore, for SVRG [26] and SARAH [38], the complexity is

$$\sum_{k=1}^{K}\mathcal{O}\left(\left(\bar{n} + \frac{\ell}{\mu_k} + 1\right)\log(n/\delta)\right) = \mathcal{O}((n + \sqrt{n}\ell D/L)\log(n/\delta)),$$

since $\sum_{k=1}^{K}1/\mu_k \leq 1/\mu \leq \mathcal{O}(\sqrt{n}D/L)$ by the settings of $\mu_k$ and $\mu$ in Algorithm 2. Similarly for Katyusha [2], we can compute that the total complexity is $\mathcal{O}((n + n^{3/4}\sqrt{\ell D/L})\log(n/\delta))$ since $\sum_{k=1}^{K}1/\sqrt{\mu_k} \leq \mathcal{O}(n^{1/4}\sqrt{D/L})$. The linear time-complexity can be achieved with fast accelerated or variance reduced algorithms, up to logarithmic factors.

# B  Differentially Private Stochastic Minimax Optimization

In this section, we give the analysis of the near-linear time algorithms for smooth DP-SMO with near-optimal utility guarantees on the (population) duality gap. We first present some useful lemmas.

## B.1  Supporting Lemmas

Here we overuse notations to provide some general properties of the duality gap for a smooth and strongly-convex–strongly-concave function $\tilde{f}(x, y)$.

**Lemma B.1.** *Let $f(x, y)$ be a $\mu_x$-strongly convex $\mu_y$-strongly concave function with one saddle point $(x^*, y^*) \in \mathcal{X} \times \mathcal{Y}$. Then for any $(\tilde{x}, \tilde{y}) \in \mathcal{X} \times \mathcal{Y}$, its duality gap satisfies that*

$$\max_{y\in\mathcal{Y}} f(\tilde{x}, y) - \min_{x\in\mathcal{X}} f(x, \tilde{y}) \geq \frac{\mu_x}{2}\|\tilde{x} - x^*\|^2 + \frac{\mu_y}{2}\|\tilde{y} - y^*\|^2.$$

*If $f(x, y)$ is also $\ell$-smooth and the saddle point $(x^*, y^*)$ is in the interior of $\mathcal{X} \times \mathcal{Y}$, it holds that*

$$\max_{y\in\mathcal{Y}} f(\tilde{x}, y) - \min_{x\in\mathcal{X}} f(x, \tilde{y}) \leq \frac{(\kappa_y + 1)\ell}{2}\|\tilde{x} - x^*\|^2 + \frac{(\kappa_x + 1)\ell}{2}\|\tilde{y} - y^*\|^2,$$

*where $\kappa_x = \ell/\mu_x$ and $\kappa_y = \ell/\mu_y$ are condition numbers.*

*Proof.* First, by optimality of $(x^*, y^*)$ and strong-convexity of $f(\cdot, y^*)$ and $-f(x^*, \cdot)$, we know

$$\max_{y \in \mathcal{Y}} f(\tilde{x}, y) - \min_{x \in \mathcal{X}} f(x, \tilde{y}) \geq f(\tilde{x}, y^*) - f(x^*, y^*) + f(x^*, y^*) - f(x^*, \tilde{y})$$

$$\geq \frac{\mu_x}{2} \|\tilde{x} - x^*\|^2 + \frac{\mu_y}{2} \|\tilde{y} - y^*\|^2.$$

The inequality also implies that for any $(\tilde{x}, \tilde{y}) \in \mathcal{X} \times \mathcal{Y}$, it holds that

$$f(\tilde{x}, y^*) - f(x^*, \tilde{y}) \geq \frac{\mu_x}{2} \|\tilde{x} - x^*\|^2 + \frac{\mu_y}{2} \|\tilde{y} - y^*\|^2. \tag{10}$$

To show the second part, we first introduce some notations. Let $\Phi(x) = \max_{y \in \mathcal{Y}} f(x, y)$ be the primal function and $\Psi(y) = \min_{x \in \mathcal{X}} f(x, y)$ be the dual function. When $f(x, y)$ is $\mu_x$-strongly convex, $\mu_y$-strongly concave and $\ell$-smooth, we have that $\Phi(x)$ is $(\kappa_y + 1)\ell$-smooth and $\Psi(y)$ is $(\kappa_x + 1)\ell$-smooth, which is a standard result in the literature (e.g., see [51, Proposition 1]). Since $(x^*, y^*)$ is the interior point, we know that

$$\max_{y \in \mathcal{Y}} f(\tilde{x}, y) - \min_{x \in \mathcal{X}} f(x, \tilde{y}) = \Phi(\tilde{x}) - f(x^*, y^*) + f(x^*, y^*) - \Psi(\tilde{y})$$

$$= \Phi(\tilde{x}) - \Phi(x^*) + \Psi(y^*) - \Psi(\tilde{y})$$

$$\leq \frac{(\kappa_y + 1)\ell}{2} \|\tilde{x} - x^*\|^2 + \frac{(\kappa_x + 1)\ell}{2} \|\tilde{y} - y^*\|^2,$$

where the last inequality follows by the smoothness of $\Phi(x)$ and $-\Psi(y)$. The same result is also obtained by Zhang et al. [51] (see proof of Theorem 3 in their appendix). $\qquad\square$

In the following, we give the stability and generalization results of empirical saddle point problems proved in Zhang et al. [51]. Lemma B.2 considers the case when $f(x, y; \xi)$ is strongly-convex–strongly-concave, and its regularized version Lemma 4.4 when $f(x, y; \xi)$ is convex-concave is in the main text. The same as the minimization case in Lemma A.2 and A.3, Lemma 4.4 does not require Lipschitzness of the regularizer and is not a trivial extension of Lemma B.2.

**Lemma B.2.** *[51, Lemma 1 and Theorem 1] Consider a stochastic minimax problem such that $f(x, y; \xi)$ is $\mu_x$-strongly convex $\mu_y$-strongly concave and L-Lipschitz w.r.t. $x \in \mathcal{X}$ and $y \in \mathcal{Y}$. Let $\mu = \min\{\mu_x, \mu_y\}$ and denote the empirical saddle point of function $\hat{F}_S(x, y)$ as $(\hat{x}_S^*, \hat{y}_S^*)$ given dataset $S$ with $n$ i.i.d. samples. Then for any neighboring datasets $S \sim S'$, we have that*

$$\mu_x \|\hat{x}_S^* - \hat{x}_{S'}^*\|^2 + \mu_y \|\hat{y}_S^* - \hat{y}_{S'}^*\|^2 \leq \frac{4L^2}{\mu n^2}.$$

*The stability result implies the generalization error of the empirical solution can be bounded as*

$$\max_{y \in \mathcal{Y}} \mathbb{E}[F(\hat{x}_S^*, y)] - \min_{x \in \mathcal{X}} \mathbb{E}[F(x, \hat{y}_S^*)] \leq \frac{2\sqrt{2}L^2}{\mu n},$$

*measured by the population weak duality gap.*

We only prove the regularized version in Lemma 4.4 for completeness. The proof of above Lemma B.2 is simpler and nearly the same, so it will not be repeated here.

*Proof of Lemma 4.4.* The same as the proof of Lemma A.3, we let neighboring datasets $S = \{\xi_1, \cdots, \xi_i, \cdots, \xi_n\}$ and $S_i' = \{\xi_1, \cdots, \xi_i', \cdots, \xi_n\}$, where $\xi_i$ and $\xi_i'$ are sampled independently.

Applying (10) in the proof of Lemma B.1 for $\hat{F}_S(x, y) + G(x, y)$ and $\hat{F}_{S_i'}(x, y) + G(x, y)$, we obtain

$$\hat{F}_S(\hat{x}_{S_i'}^*, \hat{y}_S^*) + G(\hat{x}_{S_i'}^*, \hat{y}_S^*) - \hat{F}_S(\hat{x}_S^*, \hat{y}_{S_i'}^*) - G(\hat{x}_S^*, \hat{y}_{S_i'}^*) \geq \frac{\mu_x}{2} \|\hat{x}_S^* - \hat{x}_{S_i'}^*\|^2 + \frac{\mu_y}{2} \|\hat{y}_S^* - \hat{y}_{S_i'}^*\|^2,$$

$$\hat{F}_{S_i'}(\hat{x}_S^*, \hat{y}_{S_i'}^*) + G(\hat{x}_S^*, \hat{y}_{S_i'}^*) - \hat{F}_{S_i'}(\hat{x}_{S_i'}^*, \hat{y}_S^*) - G(\hat{x}_{S_i'}^*, \hat{y}_S^*) \geq \frac{\mu_x}{2} \|\hat{x}_S^* - \hat{x}_{S_i'}^*\|^2 + \frac{\mu_y}{2} \|\hat{y}_S^* - \hat{y}_{S_i'}^*\|^2.$$

Summing up the above two inequalities, we can get

$$\mu_x \|\hat{x}_S^* - \hat{x}_{S_i'}^*\|^2 + \mu_y \|\hat{y}_S^* - \hat{y}_{S_i'}^*\|^2$$

$$\leq \hat{F}_S(\hat{x}_{S_i'}^*, \hat{y}_S^*) - \hat{F}_{S_i'}(\hat{x}_{S_i'}^*, \hat{y}_S^*) + \hat{F}_{S_i'}(\hat{x}_S^*, \hat{y}_{S_i'}^*) - \hat{F}_S(\hat{x}_S^*, \hat{y}_{S_i'}^*)$$

$$\overset{(a)}{=} \frac{1}{n}[f(\hat{x}_{S_i'}^*, \hat{y}_S^*; \xi_i) - f(\hat{x}_{S_i'}^*, \hat{y}_S^*; \xi_i')] + \frac{1}{n}[f(\hat{x}_S^*, \hat{y}_{S_i'}^*; \xi_i') - f(\hat{x}_S^*, \hat{y}_{S_i'}^*; \xi_i)]$$

$$= \frac{1}{n}[f(\hat{x}_{S_i'}^*, \hat{y}_S^*; \xi_i) - f(\hat{x}_S^*, \hat{y}_{S_i'}^*; \xi_i)] + \frac{1}{n}[f(\hat{x}_S^*, \hat{y}_{S_i'}^*; \xi_i') - f(\hat{x}_{S_i'}^*, \hat{y}_S^*; \xi_i')]$$

$$\overset{(b)}{\leq} \frac{2L}{n}\sqrt{\|\hat{x}_S^* - \hat{x}_{S_i'}^*\|^2 + \|\hat{y}_S^* - \hat{y}_{S_i'}^*\|^2}$$

$$\overset{(c)}{\leq} \frac{2L}{n\sqrt{\mu}} \cdot \sqrt{\mu_x \|\hat{x}_S^* - \hat{x}_{S_i'}^*\|^2 + \mu_y \|\hat{y}_S^* - \hat{y}_{S_i'}^*\|^2}. \tag{11}$$

where $(a)$ uses the following equation that holds for any $(x, y)$:

$$\hat{F}_S(x, y) - \hat{F}_{S_i'}(x, y) = \frac{1}{n}[f(x, y; \xi_i) - f(x, y; \xi_i')],$$

as a consequence of (4), $(b)$ is true since $f(x, y; \xi)$ is $L$-Lipschitz, i.e.,

$$|f(x_1, y_1; \xi) - f(x_2, y_2; \xi)|^2 \leq L^2 (\|x_1 - x_2\|^2 + \|y_1 - y_2\|^2), \quad \forall x_1, x_2 \in \mathcal{X}, y_1, y_2 \in \mathcal{Y},$$

and $(c)$ follows from the definition that $\mu = \min\{\mu_x, \mu_y\}$. As a result of (11), we have the following stability bound that holds for neighboring datasets $S \sim S_i'$:

$$\mu_x \|\hat{x}_S^* - \hat{x}_{S_i'}^*\|^2 + \mu_y \|\hat{y}_S^* - \hat{y}_{S_i'}^*\|^2 \leq \frac{4L^2}{\mu n^2}. \tag{12}$$

Then we can analyze the generalization error using stability results. We first bound the error of $\hat{x}_S^*$ as:

$$\max_{y \in \mathcal{Y}} \mathbb{E}\Big[F(\hat{x}_S^*, y) + G(\hat{x}_S^*, y)\Big] - \max_{y \in \mathcal{Y}} \mathbb{E}\Big[\hat{F}_S(\hat{x}_S^*, y) + G(\hat{x}_S^*, y)\Big]$$

$$\overset{(a)}{\leq} \max_{y \in \mathcal{Y}} \Big\{ \mathbb{E}[F(\hat{x}_S^*, y)] - \mathbb{E}[\hat{F}_S(\hat{x}_S^*, y)] \Big\}$$

$$\overset{(b)}{=} \max_{y \in \mathcal{Y}} \Big\{ \frac{1}{n} \sum_{i=1}^n \mathbb{E}[F(\hat{x}_{S_i'}^*, y)] - \mathbb{E}[\hat{F}_S(\hat{x}_S^*, y)] \Big\}$$

$$\overset{(c)}{=} \max_{y \in \mathcal{Y}} \frac{1}{n} \sum_{i=1}^n \mathbb{E}\Big[ f(\hat{x}_{S_i'}^*, y; \xi_i) - f(\hat{x}_S^*, y; \xi_i) \Big]$$

$$\leq \frac{L}{n} \sum_{i=1}^n \mathbb{E}\|\hat{x}_S^* - \hat{x}_{S_i'}^*\|, \tag{13}$$

where $(a)$ holds by the inequality $\max_y h_1(y) - \max_y h_2(y) \leq \max_y \{h_1(y) - h_2(y)\}$ for any function $h_1$ and $h_2$, $(b)$ holds since $x_S^*$ and $x_{S_i'}^*$ have the same distribution, and $(c)$ follows from the definition of $\hat{F}_S(x, y)$ and the fact that $S_i'$ is independent from $\xi_i$ so one can first take expectation w.r.t. $\xi_i$. Similarly for $\hat{y}_S^*$, by the inequality $\min_x h_1(x) - \min_x h_2(x) \leq \max_x \{h_1(x) - h_2(x)\}$,

$$\min_{x \in \mathcal{X}} \mathbb{E}\Big[\hat{F}_S(x, \hat{y}_S^*) + G(x, \hat{y}_S^*)\Big] - \min_{x \in \mathcal{X}} \mathbb{E}\Big[F(x, \hat{y}_S^*) + G(x, \hat{y}_S^*)\Big]$$

$$\leq \max_{x \in \mathcal{X}} \Big\{ \mathbb{E}[\hat{F}_S(x, \hat{y}_S^*)] - \mathbb{E}[F(x, \hat{y}_S^*)] \Big\}$$

$$= \max_{x \in \mathcal{X}} \frac{1}{n} \sum_{i=1}^n \mathbb{E}\Big[ f(x, \hat{y}_S^*; \xi_i) - f(x, \hat{y}_{S_i'}^*; \xi_i) \Big]$$

$$\leq \frac{L}{n} \sum_{i=1}^n \mathbb{E}\|\hat{y}_S^* - \hat{y}_{S_i'}^*\|. \tag{14}$$

Combining the above two inequalities (13) and (14), we obtain that

$$\max_{y \in \mathcal{Y}} \mathbb{E}\Big[F(\hat{x}_S^*, y) + G(\hat{x}_S^*, y)\Big] - \min_{x \in \mathcal{X}} \mathbb{E}\Big[F(x, \hat{y}_S^*) + G(x, \hat{y}_S^*)\Big]$$

$$\leq \max_{y \in \mathcal{Y}} \mathbb{E}\Big[\hat{F}_S(\hat{x}_S^*, y) + G(\hat{x}_S^*, y)\Big] - \min_{x \in \mathcal{X}} \mathbb{E}\Big[\hat{F}_S(x, \hat{y}_S^*) + G(x, \hat{y}_S^*)\Big] \qquad (15)$$

$$+ \frac{L}{n} \sum_{i=1}^{n} \mathbb{E}\Big[\|\hat{x}_S^* - \hat{x}_{S_i'}^*\| + \|\hat{y}_S^* - \hat{y}_{S_i'}^*\|\Big].$$

The first two terms in the RHS of (15) can be bounded by Cauchy–Schwarz inequality and the optimality of $(\hat{x}_S^*, \hat{y}_S^*)$,

$$\max_{y \in \mathcal{Y}} \mathbb{E}\Big[\hat{F}_S(\hat{x}_S^*, y) + G(\hat{x}_S^*, y)\Big] - \min_{x \in \mathcal{X}} \mathbb{E}\Big[\hat{F}_S(x, \hat{y}_S^*) + G(x, \hat{y}_S^*)\Big]$$

$$\leq \mathbb{E}\Big[\max_{y \in \mathcal{Y}}\big\{\hat{F}_S(\hat{x}_S^*, y) + G(\hat{x}_S^*, y)\big\} - \min_{x \in \mathcal{X}}\big\{\hat{F}_S(x, \hat{y}_S^*) + G(x, \hat{y}_S^*)\big\}\Big]$$

$$= 0.$$

For the last term in the RHS of (15), we have that

$$\Big(\|\hat{x}_S^* - \hat{x}_{S_i'}^*\| + \|\hat{y}_S^* - \hat{y}_{S_i'}^*\|\Big)^2 \leq 2\|\hat{x}_S^* - \hat{x}_{S_i'}^*\|^2 + 2\|\hat{y}_S^* - \hat{y}_{S_i'}^*\|^2$$

$$\leq \frac{2}{\mu}\Big(\mu_x\|\hat{x}_S^* - \hat{x}_{S_i'}^*\|^2 + \mu_y\|\hat{y}_S^* - \hat{y}_{S_i'}^*\|^2\Big)$$

$$\leq \frac{8L^2}{\mu^2 n^2}, \qquad (16)$$

where the last inequality follows from the stability (12). Then plugging the above two bounds back into (15), we obtain that

$$\max_{y \in \mathcal{Y}} \mathbb{E}\Big[F(\hat{x}_S^*, y) + G(\hat{x}_S^*, y)\Big] - \min_{x \in \mathcal{X}} \mathbb{E}\Big[F(x, \hat{y}_S^*) + G(x, \hat{y}_S^*)\Big] \leq \frac{2\sqrt{2}L^2}{\mu n},$$

which is the generalization error of $(\hat{x}_S^*, \hat{y}_S^*)$ measured by the population weak duality gap. $\qquad \square$

In the analysis of the phased algorithms for convex-concave functions, we will use the following corollary of Lemma 4.4.

**Corollary B.3.** *Under the same settings as Lemma 4.4. Let $(u_S, v_S) \in \mathcal{X} \times \mathcal{Y}$ be some points that may have dependence on the dataset $S$. Suppose their stability w.r.t. $S$ is bounded as $\mathbb{E}\|u_S - u_{S'}\| \leq \Delta_u$ and $\mathbb{E}\|v_S - v_{S'}\| \leq \Delta_v$ for any neighboring datasets $S \sim S'$. Then it holds that*

$$\mathbb{E}[F(\hat{x}_S^*, v_S) - F(u_S, \hat{y}_S^*)] \leq \mathbb{E}[G(u_S, \hat{y}_S^*) - G(\hat{x}_S^*, v_S)] + \frac{2\sqrt{2}L^2}{\mu n} + L(\Delta_u + \Delta_v).$$

*As a special case, when $u$ and $v$ are independent of $S$, the third term vanishes since $\Delta_u = \Delta_v = 0$.*

*Proof.* When $v_S$ has dependence on $S$, by the same reason as (13), we have

$$\mathbb{E}\Big[F(\hat{x}_S^*, v_S) + G(\hat{x}_S^*, v_S)\Big] - \mathbb{E}\Big[\hat{F}_S(\hat{x}_S^*, v_S) + G(\hat{x}_S^*, v_S)\Big]$$

$$= \frac{1}{n} \sum_{i=1}^{n} \mathbb{E}[F(\hat{x}_{S_i'}^*, v_{S_i'})] - \mathbb{E}[\hat{F}_S(\hat{x}_S^*, v_S)]$$

$$= \frac{1}{n} \sum_{i=1}^{n} \mathbb{E}\Big[f(\hat{x}_{S_i'}^*, v_{S_i'}; \xi_i) - f(\hat{x}_S^*, v_S; \xi_i)\Big]$$

$$\leq \frac{L}{n} \sum_{i=1}^{n} \mathbb{E}\Big[\|\hat{x}_{S_i'}^* - \hat{x}_S^*\| + \|v_{S_i'} - v_S\|\Big],$$

and similarly by (14), we get

$$\mathbb{E}\Big[\hat{F}_S(u_S, \hat{y}_S^*) + G(u_S, \hat{y}_S^*)\Big] - \mathbb{E}\Big[F(u_S, \hat{y}_S^*) + G(u_S, \hat{y}_S^*)\Big] \leq \frac{L}{n}\sum_{i=1}^{n}\mathbb{E}\Big[\|\hat{y}_{S_i'}^* - \hat{y}_S^*\| + \|u_{S_i'} - u_S\|\Big].$$

Moreover, since $(\hat{x}_S^*, \hat{y}_S^*)$ is the saddle point of $\hat{F}_S(x,y) + G(x,y)$, we have

$$
\begin{aligned}
\hat{F}_S(\hat{x}_S^*, v_S) &+ G(\hat{x}_S^*, v_S) - \hat{F}_S(u_S, \hat{y}_S^*) - G(u_S, \hat{y}_S^*) \\
&\leq \hat{F}_S(\hat{x}_S^*, \hat{y}_S^*) + G(\hat{x}_S^*, \hat{y}_S^*) - \hat{F}_S(\hat{x}_S^*, \hat{y}_S^*) - G(\hat{x}_S^*, \hat{y}_S^*) \\
&= 0.
\end{aligned}
$$

As a result, by the three inequalities above, we have

$$
\begin{aligned}
\mathbb{E}\Big[F(\hat{x}_S^*, v_S) &+ G(\hat{x}_S^*, v_S)\Big] - \mathbb{E}\Big[F(u_S, \hat{y}_S^*) + G(u_S, \hat{y}_S^*)\Big] \\
&\leq \mathbb{E}\Big[\hat{F}_S(\hat{x}_S^*, v_S) + G(\hat{x}_S^*, v_S)\Big] - \mathbb{E}\Big[\hat{F}_S(u_S, \hat{y}_S^*) + G(u_S, \hat{y}_S^*)\Big] \\
&\quad + \frac{L}{n}\sum_{i=1}^{n}\mathbb{E}\Big[\|\hat{x}_S^* - \hat{x}_{S_i'}^*\| + \|\hat{y}_S^* - \hat{y}_{S_i'}^*\|\Big] + \frac{L}{n}\sum_{i=1}^{n}\mathbb{E}\Big[\|u_S - u_{S_i'}\| + \|v_S - v_{S_i'}\|\Big] \\
&\leq \frac{2\sqrt{2}L^2}{\mu n} + L(\Delta_u + \Delta_v),
\end{aligned}
$$

where the last inequality follows from (16) and stability of $u_S, v_S$. □

## B.2  Near-Linear Time Algorithms for Smooth Strongly-Convex–Strongly-Concave Functions

Algorithm 3 achieves near-optimal utility guarantees on the strong duality gap for SC-SC functions with near-linear time-complexity. Its analysis is provided in the proof of Theorem 4.3 below.

*Proof of Theorem 4.3.*  For the privacy guarantee, we first bound the sensitivity of algorithm $\mathcal{A}$. Given neighboring datasets $S \sim S'$, by Lemma B.2, for $\mu = \min\{\mu_x, \mu_y\}$, we know that

$$
\begin{aligned}
\|\hat{x}_S^* - \hat{x}_{S'}^*\| &\leq \sqrt{\frac{1}{\mu_x}(\mu_x\|\hat{x}_S^* - \hat{x}_{S'}^*\|^2 + \mu_y\|\hat{y}_S^* - \hat{y}_{S'}^*\|^2)} \\
&\leq \frac{2L}{n}\sqrt{\frac{1}{\mu_x\mu}}.
\end{aligned}
$$

For the same reason, by the guarantee of $\mathcal{A}$ in Algorithm 3, we have $\|A_x(S) - \hat{x}_S^*\| \leq (L/n)\sqrt{1/(\mu_x\mu)}$ with probability at least $1 - \delta/8$. Thus the sensitivity of $\mathcal{A}_x$ is bounded as

$$
\begin{aligned}
\|A_x(S) - A_x(S')\| &\leq \|A_x(S) - \hat{x}_S^*\| + \|\hat{x}_S^* - \hat{x}_{S'}^*\| + \|\hat{x}_{S'}^* - A_x(S')\| \\
&\leq \frac{4L}{n}\sqrt{\frac{1}{\mu_x\mu}},
\end{aligned}
\tag{17}
$$

with probability at least $1 - \delta/4$ by the union bound. Applying Gaussian mechanism in Definition 2, $\mathcal{A}_x + \mathcal{N}(0, \sigma_x^2 I_{d_x})$ is $(\varepsilon/2, \delta/2)$-DP when setting $\sigma_x = (8L/(n\varepsilon))\sqrt{2\log(5/\delta)/(\mu_x\mu)}$. Similarly we can bound the sensitivity of $\mathcal{A}_y$ and the setting of $\sigma_y$ in Algorithm 3 guarantees $(\varepsilon/2, \delta/2)$-DP of $\mathcal{A}_y + \mathcal{N}(0, \sigma_y^2 I_{d_y})$. Finally by the basic composition in Lemma 2.1, Algorithm 3 is $(\varepsilon, \delta)$-DP.

Next, we prove the utility guarantees for the output $(\tilde{x}, \tilde{y})$. For the empirical bound, we apply Lemma B.1 for $\hat{F}_S(x,y)$ under the assumption that $(\hat{x}_S^*, \hat{y}_S^*)$ is the interior point and get

$$
\begin{aligned}
\mathbb{E}\Big[\max_{y\in\mathcal{Y}}\hat{F}_S(\tilde{x}, y) - \min_{x\in\mathcal{X}}\hat{F}_S(x, \tilde{y})\Big] &\leq \frac{(\kappa_y + 1)\ell}{2}\mathbb{E}\|\tilde{x} - \hat{x}_S^*\|^2 + \frac{(\kappa_x + 1)\ell}{2}\mathbb{E}\|\tilde{y} - \hat{y}_S^*\|^2 \\
&\leq (\kappa_y + 1)\ell \cdot \mathbb{E}\big[\|\tilde{x} - \mathcal{A}_x(S)\|^2 + \|\mathcal{A}_x(S) - \hat{x}_S^*\|^2\big] \\
&\quad + (\kappa_x + 1)\ell \cdot \mathbb{E}\big[\|\tilde{y} - \mathcal{A}_y(S)\|^2 + \|\mathcal{A}_y(S) - \hat{y}_S^*\|^2\big].
\end{aligned}
\tag{18}
$$

According to the values of $\sigma_x$ and $\sigma_y$ in Algorithm 3 to guarantee $(\varepsilon, \delta)$-DP, we know

$$
\begin{aligned}
(\kappa_y + 1)\ell \cdot \mathbb{E}\|\tilde{x} - \mathcal{A}_x(S)\|^2 &+ (\kappa_x + 1)\ell \cdot \mathbb{E}\|\tilde{y} - \mathcal{A}_y(S)\|^2 \\
&\leq (\kappa_x \kappa_y + \kappa) \cdot \mathbb{E}\big[\mu_x\|\tilde{x} - \mathcal{A}_x(S)\|^2 + \mu_y\|\tilde{y} - \mathcal{A}_y(S)\|^2\big] \\
&\leq 256 L^2(\kappa_x \kappa_y + \kappa)\frac{d\log(5/\delta)}{\mu n^2 \varepsilon^2},
\end{aligned}
\tag{19}
$$

where we let $\mu = \min\{\mu_x, \mu_y\}$, $\kappa = \ell/\mu$ and $d = \max\{d_x, d_y\}$. Similarly by the guarantee of the algorithm $\mathcal{A}$ in Remark 3, we have

$$
\begin{aligned}
(\kappa_y + 1)\ell \cdot \mathbb{E}\|\mathcal{A}_x(S) - \hat{x}_S^*\|^2 &+ (\kappa_x + 1)\ell \cdot \mathbb{E}\|\mathcal{A}_y(S) - \hat{y}_S^*\|^2 \\
&\leq (\kappa_x \kappa_y + \kappa) \cdot \mathbb{E}\big[\mu_x\|\mathcal{A}_x(S) - \hat{x}_S^*\|^2 + \mu_y\|\mathcal{A}_y(S) - \hat{y}_S^*\|^2\big] \\
&\leq L^2(\kappa_x \kappa_y + \kappa)\frac{\delta}{8\mu n^2}.
\end{aligned}
\tag{20}
$$

As a result of above two inequalities and (18), when $d \geq 1$, $\varepsilon < 1$ and $\delta < 1/n$, we have that

$$
\mathbb{E}\left[\max_{y \in \mathcal{Y}} \hat{F}_S(\tilde{x}, y) - \min_{x \in \mathcal{X}} \hat{F}_S(x, \tilde{y})\right] \leq 257 L^2(\kappa_x \kappa_y + \kappa)\frac{d\log(5/\delta)}{\mu n^2 \varepsilon^2}.
$$

Similarly, we can apply Lemma B.1 for $F(x, y)$ to obtain the population guarantee as

$$
\begin{aligned}
\mathbb{E}\left[\max_{y \in \mathcal{Y}} F(\tilde{x}, y) - \min_{x \in \mathcal{X}} F(x, \tilde{y})\right] &\leq \frac{(\kappa_y + 1)\ell}{2}\mathbb{E}\|\tilde{x} - x^*\|^2 + \frac{(\kappa_x + 1)\ell}{2}\mathbb{E}\|\tilde{y} - y^*\|^2) \\
&\leq \frac{3(\kappa_y + 1)\ell}{2}\mathbb{E}\big[\|\tilde{x} - \mathcal{A}_x(S)\|^2 + \|\mathcal{A}_x(S) - \hat{x}_S^*\|^2 + \|\hat{x}_S^* - x^*\|^2\big] \\
&\quad + \frac{3(\kappa_x + 1)\ell}{2}\mathbb{E}\big[\|\tilde{y} - \mathcal{A}_y(S)\|^2 + \|\mathcal{A}_y(S) - \hat{y}_S^*\|^2 + \|\hat{y}_S^* - y^*\|^2\big] \\
&\leq L^2(\kappa_x \kappa_y + \kappa)\left(\frac{384 d\log(5/\delta)}{\mu n^2 \varepsilon^2} + \frac{3\delta}{16\mu n^2} + \frac{6\sqrt{2}}{\mu n}\right) \\
&< 3L^2(\kappa_x \kappa_y + \kappa)\left(\frac{3}{\mu n} + \frac{128 d\log(5/\delta)}{\mu n^2 \varepsilon^2}\right),
\end{aligned}
$$

where we use (19) and (20), and the bound for the distance between $(\hat{x}_S^*, \hat{y}_S^*)$ and $(x^*, y^*)$ follows from Lemma B.1 and the generalization results in Lemma B.2 such that

$$
\mathbb{E}\left[\frac{\mu_x}{2}\|\hat{x}_S^* - x^*\|^2 + \frac{\mu_y}{2}\|\hat{y}_S^* - y^*\|^2\right] \leq \frac{2\sqrt{2}L^2}{\mu n},
$$

where $\mu = \min\{\mu_x, \mu_y\}$ simplifies the notations. We conclude the proof by a remark. When the saddle points are not interior points, we can instead obtain the guarantee on the distance between $(\tilde{x}, \tilde{y})$ and $(\hat{x}_S^*, \hat{y}_S^*)$ or $(x^*, y^*)$, which is weaker than the duality gap. $\qquad\square$

**Near-optimality:** The utility bound on the strong duality gap of our algorithm scales with $\mathcal{O}\big(\kappa^2\big(1/(\mu n) + d\log(1/\delta)/(\mu n^2 \varepsilon^2)\big)\big)$. Since the SC-SC minimax problem can be regarded as a strongly-convex minimization problem when restricting the domain $\mathcal{Y}$ as a singleton, the lower-bound [6, 7] $\Omega(1/(\mu_x n) + d_x \log(1/\delta)/(\mu_x n^2 \varepsilon^2))$ of a $\mu_x$-strongly convex DP-SCO problem can be regarded as a trivial lower-bound of DP-SMO problem. Therefore, our rate is optimal w.r.t. $n$ and $(\varepsilon, \delta)$, but it remains unclear whether the dependence w.r.t. $\mu_x$, $\mu_y$ and $d_x$, $d_y$ can be further improved as the exact lower-bound for SC-SC DP-SMO is still an open question. Nonetheless, we achieve near-optimal bound when the problem is not ill-conditioned.

**Gradient Complexity:** Algorithm 3 requires that $\mu_x\|\mathcal{A}_x(S) - \hat{x}_S^*\|^2 + \mu_y\|\mathcal{A}_y(S) - \hat{y}_S^*\|^2 \leq L^2/(\mu n^2)$ holds with probability at least $1 - \delta/8$. Existing methods for smooth SC-SC finite-sum saddle point problems guarantee that $\mathbb{E}[\max_{y \in \mathcal{Y}} \hat{F}_S(\mathcal{A}_x(S), y) - \min_{x \in \mathcal{X}} \hat{F}_S(x, \mathcal{A}_y(S))] \leq \gamma$ with $\mathcal{O}(T(n, \kappa_x, \kappa_y)\log(1/\gamma))$ gradient complexity. For example, Extragradient [43] uses $\mathcal{O}(n\kappa \log(1/\gamma))$ gradient queries and SVRG/SAGA [39] needs $\mathcal{O}((n + \kappa^2)\log(1/\gamma))$ gradient

computations, where $\kappa = \max\{\kappa_x, \kappa_y\}$. Since $\mathbb{E}[\mu_x \|\mathcal{A}_x(S) - \hat{x}_S^*\|^2 + \mu_y \|\mathcal{A}_y(S) - \hat{y}_S^*\|^2] \leq 2\gamma$ by Lemma B.1, we can then apply Markov's inequality and obtain that

$$\mathbb{P}\left( \left( \mu_x \|\mathcal{A}_x(S) - \hat{x}_S^*\|^2 + \mu_y \|\mathcal{A}_y(S) - \hat{y}_S^*\|^2 \right) \geq \frac{L^2}{\mu n^2} \right) \leq \frac{2\gamma\mu n^2}{L^2}.$$

Setting $\gamma = \delta L^2/(16\mu n^2)$, the RHS becomes $\delta/8$ and the requirement of $\mathcal{A}$ is satisfied. This implies the complexity of Algorithm 3 is $\mathcal{O}(T(n, \kappa_x, \kappa_y) \log(n/\delta))$ as discussed in Remark 3. Therefore, we achieve the linear-time gradient complexity up to logarithmic factors.

## B.3 Near-Linear Time Algorithms for Smooth Convex-Concave Functions

In this section, we analyze the algorithm for smooth convex-concave DP-SMO. We first present a lemma that extends the results that the proximal operator is non-expansive to minimax settings.

**Lemma B.4.** *Let $f(x, y)$ be a convex-concave function on the closed convex domain $\mathcal{X} \times \mathcal{Y}$. For some $(u, v) \in \mathcal{X} \times \mathcal{Y}$ and $\mu_x, \mu_y > 0$, we define*

$$F_{u,v}(x, y) \triangleq f(x, y) + \frac{\mu_x}{2}\|x - u\|^2 - \frac{\mu_y}{2}\|y - v\|^2.$$

*Denote the saddle point of $F_{u,v}(x, y)$ as $(x_{u,v}^*, y_{u,v}^*) \in \mathcal{X} \times \mathcal{Y}$. Then it holds that*

$$\mu_x \|x_{u,v}^* - x_{u',v'}^*\|^2 + \mu_y \|y_{u,v}^* - y_{u',v'}^*\|^2 \leq \mu_x \|u - u'\|^2 + \mu_y \|v - v'\|^2,$$

*where $(x_{u',v'}^*, y_{u',v'}^*) \in \mathcal{X} \times \mathcal{Y}$ is the saddle point of $F_{u',v'}(x, y)$ defined in the same way as $F_{u,v}(x, y)$ for some point $(u', v') \in \mathcal{X} \times \mathcal{Y}$.*

*Proof.* For $(x, y) \in \mathbb{R}^{d_x} \times \mathbb{R}^{d_y}$, we define $f_{\mathcal{X},\mathcal{Y}}(x, y) := f(x, y) + \mathcal{I}_{\mathcal{X}}(x) - \mathcal{I}_{\mathcal{Y}}(y)$, where $\mathcal{I}_C$ is the indicator function of some set $C$, i.e., $\mathcal{I}_C(x) = 0$ if $x \in C$ and $\mathcal{I}_C(x) = \infty$ otherwise. Since $\mathcal{I}_{\mathcal{X}}$ and $\mathcal{I}_{\mathcal{Y}}$ are both convex when the domain is convex, $f_{\mathcal{X},\mathcal{Y}}(x, y)$ is convex-concave. By the above definition, $(x_{u,v}^*, y_{u,v}^*)$ is the saddle point of $\bar{F}_{u,v}(x, y) := f_{\mathcal{X},\mathcal{Y}}(x, y) + (\mu_x/2)\|x - u\|^2 - (\mu_y/2)\|y - v\|^2$. Applying the optimality condition, i.e., $0 \in \partial \bar{F}_{u,v}(x_{u,v}^*, y_{u,v}^*)$, we know that

$$\mu_x(u - x_{u,v}^*) \in \partial_x f_{\mathcal{X},\mathcal{Y}}(x_{u,v}^*, y_{u,v}^*), \quad \mu_y(v - y_{u,v}^*) \in -\partial_y f_{\mathcal{X},\mathcal{Y}}(x_{u,v}^*, y_{u,v}^*), \quad (21)$$

where $\partial_x f_{\mathcal{X},\mathcal{Y}}$ and $\partial_y f_{\mathcal{X},\mathcal{Y}}$ are partial subgradients. Similarly for $(x_{u',v'}^*, y_{u',v'}^*)$ and $\bar{F}_{u',v'}$, we have

$$\mu_x(u' - x_{u',v'}^*) \in \partial_x f_{\mathcal{X},\mathcal{Y}}(x_{u',v'}^*, y_{u',v'}^*), \quad \mu_y(v' - y_{u',v'}^*) \in -\partial_y f_{\mathcal{X},\mathcal{Y}}(x_{u',v'}^*, y_{u',v'}^*). \quad (22)$$

By the property that $(\partial_x f_{\mathcal{X},\mathcal{Y}}, -\partial_y f_{\mathcal{X},\mathcal{Y}})$ is a monotone operator when $f_{\mathcal{X},\mathcal{Y}}(x, y)$ is convex-concave (see [40, Theorem 1] or [19, Lemma 1]), it holds for any $x_1, x_2 \in \mathbb{R}^{d_x}$, $y_1, y_2 \in \mathbb{R}^{d_y}$, and subgradients $(g_x, -g_y) \in (\partial_x f_{\mathcal{X},\mathcal{Y}}, -\partial_y f_{\mathcal{X},\mathcal{Y}})$ that

$$\left(g_x(x_1, y_1) - g_x(x_2, y_2)\right)^\top (x_1 - x_2) - \left(g_y(x_1, y_1) - g_y(x_2, y_2)\right)^\top (y_1 - y_2) \geq 0.$$

Therefore, using the set membership relations (21) and (22), we get that

$$\mu_x \left((u - u') - (x_{u,v}^* - x_{u',v'}^*)\right)^\top \left(x_{u,v}^* - x_{u',v'}^*\right) + \mu_y \left((v - v') - (y_{u,v}^* - y_{u',v'}^*)\right)^\top \left(y_{u,v}^* - y_{u',v'}^*\right) \geq 0.$$

Hence, we obtain that

$$\begin{aligned}
&\mu_x \|x_{u,v}^* - x_{u',v'}^*\|^2 + \mu_y \|y_{u,v}^* - y_{u',v'}^*\|^2 \\
&\leq \mu_x(u - u')^\top (x_{u,v}^* - x_{u',v'}^*) + \mu_y(v - v')^\top (y_{u,v}^* - y_{u',v'}^*) \\
&\leq \mu_x \|u - u'\| \|x_{u,v}^* - x_{u',v'}^*\| + \mu_y \|v - v'\| \|y_{u,v}^* - y_{u',v'}^*\| \\
&\leq \sqrt{\mu_x \|x_{u,v}^* - x_{u',v'}^*\|^2 + \mu_y \|y_{u,v}^* - y_{u',v'}^*\|^2} \cdot \sqrt{\mu_x \|u - u'\|^2 + \mu_y \|v - v'\|^2},
\end{aligned}$$

by Cauchy-Schwarz inequality. The proof is thus complete. $\square$

The following lemma is a consequence of Corollary B.3 and Lemma B.4.

**Lemma B.5.** *For $k = 1, \cdots, K$, by the settings and notations in Algorithm 4, we have that*

$$\mathbb{E}[F(\hat{x}_k^*, \hat{y}_{k+1}^*) - F(\hat{x}_{k-1}^*, \hat{y}_k^*)] \leq \frac{\mu_k}{2} \mathbb{E}\|\hat{x}_{k-1}^* - \tilde{x}_{k-1}\|^2 + \frac{8L^2}{\mu\bar{n}} + \frac{\mu}{2}\mathbb{E}[\|\hat{y}_{k+1}^*\|^2 - \|\hat{y}_k^*\|^2],$$

*where $(\hat{x}_k^*, \hat{y}_k^*)$ is the saddle point of the regularized empirical function $\hat{F}_k(x,y)$ for $1 \leq k \leq K$, and $\hat{x}_0^* = x^*$, $\hat{y}_{K+1}^* \in \arg\max_{y \in \mathcal{Y}} \mathbb{E}[F(\tilde{x}_K, y)]$ are used later in the proof of Theorem 4.5.*

*Proof.* Applying Corollary B.3 for $\hat{F}_k(x,y)$ with regularization term $(\mu_k/2)\|x - \tilde{x}_{k-1}\|^2 - (\mu/2)\|y\|^2$ and dataset $S_k := \{\xi_i\}_{i=(k-1)\bar{n}+1}^{k\bar{n}}$, we have that for any $x \in \mathcal{X}$ and $y \in \mathcal{Y}$,

$$\mathbb{E}[F(\hat{x}_k^*, y) - F(x, \hat{y}_k^*)] \leq \frac{\mu_k}{2}\mathbb{E}\|x - \tilde{x}_{k-1}\|^2 - \frac{\mu_k}{2}\mathbb{E}\|\hat{x}_k^* - \tilde{x}_{k-1}\|^2 + \frac{\mu}{2}\mathbb{E}\|y\|^2 - \frac{\mu}{2}\mathbb{E}\|\hat{y}_k^*\|^2$$
$$+ \frac{2\sqrt{2}L^2}{\mu\bar{n}} + L(\Delta_x + \Delta_y), \tag{23}$$

where $\Delta_x$ and $\Delta_y$ are the stability bounds of $x$ and $y$ w.r.t. $S_k$. Let $x = \hat{x}_{k-1}^*$ and $y = \hat{y}_{k+1}^*$ in the above inequality. Note that $\hat{x}_{k-1}^*$ is independent of $S_k$ and then $\Delta_x = 0$ for $1 \leq k \leq K$. For $\Delta_y$, we need to compute the difference of $\hat{y}_{k+1}^*$ given neighboring datasets $S_k \sim S_{k'}$. In the following analysis, we denote the saddle point as $(\hat{x}_{k'}^*, \hat{y}_{k'}^*)$, the output of $\mathcal{A}$ as $(x_{k'}, y_{k'})$ and the perturbed output as $\tilde{x}_{k'}$ corresponding to the dataset $S_{k'}$. When $k \leq K - 1$, since $\hat{y}_{k+1}^*$ is the saddle point of

$$\frac{1}{\bar{n}} \sum_{i \in S_{k+1}} f(x, y; \xi_i) + \frac{\mu_{k+1}}{2}\|x - \tilde{x}_k\|^2 - \frac{\mu}{2}\|y\|^2,$$

and $\hat{y}_{k'+1}^*$ is the saddle point of

$$\frac{1}{\bar{n}} \sum_{i \in S_{k+1}} f(x, y; \xi_i) + \frac{\mu_{k+1}}{2}\|x - \tilde{x}_{k'}\|^2 - \frac{\mu}{2}\|y\|^2,$$

we can conclude from Lemma B.4 that

$$\begin{aligned}
\mu\mathbb{E}\|\hat{y}_{k+1}^* - \hat{y}_{k'+1}^*\|^2 &\leq \mu_{k+1}\mathbb{E}\|\tilde{x}_k - \tilde{x}_{k'}\|^2 \\
&= \mu_{k+1}\mathbb{E}\|x_k - x_{k'}\|^2 \\
&\leq 3\mu_{k+1}\left(\mathbb{E}\|x_k - \hat{x}_k^*\|^2 + \mathbb{E}\|\hat{x}_k^* - \hat{x}_{k'}^*\|^2 + \mathbb{E}\|x_{k'} - \hat{x}_{k'}^*\|^2\right) \\
&\leq \frac{\mu_{k+1}}{\mu_k}\left(\frac{12L^2}{\mu\bar{n}^2} + \frac{3\delta L^2}{4\mu\bar{n}^2}\right) \\
&\leq \frac{26L^2}{\mu\bar{n}^2},
\end{aligned}$$

where the equality holds since the only difference is due to the neighboring datasets $S_k \sim S_{k'}$ and the third inequality follows from Lemma 4.4 and guarantees of $\mathcal{A}$ in Remark 3. Therefore we obtain that $\Delta_y = \sqrt{26}L/(\mu\bar{n})$ for $\hat{y}_{k+1}^*$ when $k \leq K - 1$. When $k = K$, by the definition that $\hat{y}_{K+1}^* \in \arg\max_{y \in \mathcal{Y}} \mathbb{E}[F(\tilde{x}_K, y)]$, we know that $\Delta_y = 0$ since $\hat{y}_{K+1}^*$ is independent of $S$. Then by (23), it holds for all $k = 1, \cdots, K$ that

$$\mathbb{E}[F(\hat{x}_k^*, \hat{y}_{k+1}^*) - F(\hat{x}_{k-1}^*, \hat{y}_k^*)] \leq \frac{\mu_k}{2}\mathbb{E}\|\hat{x}_{k-1}^* - \tilde{x}_{k-1}\|^2 + \frac{\mu}{2}\mathbb{E}[\|\hat{y}_{k+1}^*\|^2 - \|\hat{y}_k^*\|^2] + \frac{8L^2}{\mu\bar{n}}$$

since $\|\hat{x}_k^* - \tilde{x}_{k-1}\|^2 \geq 0$ and $2\sqrt{2} + \sqrt{26} < 8$. □

With Lemma B.5, we are ready to give the proof of Theorem 4.5.

*Proof of Theorem 4.5.* The population function $F(x,y)$ has at least one saddle point on the domain $\mathcal{X} \times \mathcal{Y}$, and we denote it as $(x^*, y^*)$. First, we show that Algorithm 4 is $(\varepsilon/2, \delta/2)$-DP and give the utility bound of its output $\tilde{x}_K$. Similar to the proof of Theorem 4.3, we can obtain guarantees of each phase in Algorithm 4. By Lemma 4.4, since $\min\{\mu_k, \mu\} = \mu$, we know that the empirical solution $\hat{x}_k^*$ has stability $2L/(\bar{n}\sqrt{\mu_k\mu})$. With the guarantee of the algorithm $\mathcal{A}$ and the same statement as (17),

the sensitivity of $x_k$ is bounded by $4L/(\bar{n}\sqrt{\mu_k\mu})$ with probability $1 - \delta/4$, and thus the values of $\sigma_k$ guarantee $(\varepsilon/2, \delta/2)$-DP. As a result, by (19) and (20),

$$\mathbb{E}\|\tilde{x}_k - \hat{x}_k^*\|^2 \leq 2\mathbb{E}\|\tilde{x}_k - x_k\|^2 + 2\mathbb{E}\|x_k - \hat{x}_k^*\|^2$$

$$\leq 2d_x\sigma_k^2 + \frac{L^2}{4\mu_k}\frac{\delta}{\mu\bar{n}^2}$$

$$\leq \frac{257L^2}{\mu_k}\frac{d_x \log(5/\delta)}{\mu\bar{n}^2\varepsilon^2}. \tag{24}$$

Then we analyze the full algorithm. By the parallel composition in Lemma A.1, Algorithm 4 is $(\varepsilon/2, \delta/2)$-DP since we use disjoint datasets for different phases and each phase is $(\varepsilon/2, \delta/2)$-DP. For the utility bound of the output $\tilde{x}_K$, we start with the following decomposition:

$$\max_{y\in\mathcal{Y}} \mathbb{E}[F(\tilde{x}_K, y)] - \mathbb{E}[F(x^*, \hat{y}_1^*)] = \mathbb{E}[F(\tilde{x}_K, \hat{y}_{K+1}^*) - F(\hat{x}_K^*, \hat{y}_{K+1}^*)]$$

$$+ \sum_{k=1}^{K} \mathbb{E}\Big[F(\hat{x}_k^*, \hat{y}_{k+1}^*) - F(\hat{x}_{k-1}^*, \hat{y}_k^*)\Big], \tag{25}$$

where we let $\hat{x}_0^* = x^*$ for the saddle point of $F(x, y)$ and $\hat{y}_{K+1}^* \in \arg\max_{y\in\mathcal{Y}} \mathbb{E}[F(\tilde{x}_K, y)]$ to simplify the analysis. The first term in the RHS of (25) can be bounded by Lipschitzness of $F(x, y)$:

$$\mathbb{E}[F(\tilde{x}_K, \hat{y}_{K+1}^*) - F(\hat{x}_K^*, \hat{y}_{K+1}^*)] \leq L\sqrt{\mathbb{E}\|\tilde{x}_K - \hat{x}_K^*\|^2}$$

$$< \frac{17L^2}{\sqrt{\mu_K\mu}}\frac{\sqrt{d_x \log(5/\delta)}}{\bar{n}\varepsilon}$$

$$= \frac{17L^2}{\mu\sqrt{n}}\frac{\sqrt{d_x \log(5/\delta)}}{\bar{n}\varepsilon}$$

$$\leq 17LD \cdot \frac{\sqrt{d \log(5/\delta)}}{2\bar{n}\varepsilon}, \tag{26}$$

where the second inequality uses the guarantee of phase $K$ in (24), the equality is due to the setting that $\mu_K = \mu n$ and the last inequality follows from the choice that $\mu \geq 2L/(D\sqrt{n})$ and $d_x \leq d$. Therefore, with Lemma B.5 to bound the second term in the RHS of (25), we obtain that

$$\max_{y\in\mathcal{Y}} \mathbb{E}[F(\tilde{x}_K, y)] - \mathbb{E}[F(x^*, \hat{y}_1^*)] \leq 17LD \cdot \frac{\sqrt{d \log(5/\delta)}}{2\bar{n}\varepsilon} + \sum_{k=1}^{K}\left(\frac{\mu_k}{2}\mathbb{E}\|\hat{x}_{k-1}^* - \tilde{x}_{k-1}\|^2 + \frac{8L^2}{\mu\bar{n}}\right)$$

$$+ \frac{\mu}{2}\sum_{k=1}^{K}\mathbb{E}[\|\hat{y}_{k+1}^*\|^2 - \|\hat{y}_k^*\|^2]$$

$$\leq 17LD \cdot \frac{\sqrt{d \log(5/\delta)}}{2\bar{n}\varepsilon} + \sum_{k=2}^{K}\frac{257\mu_k}{2\mu_{k-1}}\frac{L^2 d_x \log(5/\delta)}{\mu\bar{n}^2\varepsilon^2}$$

$$+ \sum_{k=1}^{K}\frac{8L^2}{\mu\bar{n}} + \frac{\mu_1}{2}\|x^* - x_0\|^2 + \frac{\mu}{2}\|\hat{y}_{K+1}^*\|^2$$

$$\leq 4LDK^2\left(\frac{1}{\sqrt{n}} + \frac{5\sqrt{d \log(5/\delta)}}{n\varepsilon}\right) + \frac{\mu}{2}(2\|x^* - x_0\|^2 + \|\hat{y}_{K+1}^*\|^2)$$

$$\leq 8LDK^2\left(\frac{1}{\sqrt{n}} + \frac{5\sqrt{d \log(5/\delta)}}{n\varepsilon}\right), \tag{27}$$

where the second inequality is due to the guarantees of $\tilde{x}_{k-1}$ in (24) for $k \geq 2$ and the settings that $\tilde{x}_0 = x_0$, $\hat{x}_0^* = x^*$, the third inequality holds by the choice that $\mu_k = \mu \cdot 2^k$, $\mu = (L/D)\max\{2/\sqrt{n}, 13\log(n)\sqrt{d \log(5/\delta)}/(n\varepsilon)\}$, and the last inequality follows from the assumptions that $\|x^*\|^2 \leq D^2$ and $\|\hat{y}_{K+1}^*\|^2 \leq D^2$ when the initialization is $x_0 = 0$. Since $(x^*, y^*)$

---

**Algorithm 5** Phased Output Perturbation for Convex-Concave Minimax Problems

---

**Input:** Dataset $S = \{\xi_i\}_{i=1}^n$, algorithm $\mathcal{A}$, DP Parameters $(\varepsilon, \delta)$, regularizer $\mu$, initializer $y_0$.

1: Set $K = \log(n)$, $\bar{n} = n/K$ and $\tilde{y}_0 = y_0$.
2: **for** $k = 1, \cdots, K$ **do**
3:     Set $\mu_k = \mu \cdot 2^k$.
4:     Run the algorithm $\mathcal{A}$ on the smooth SC-SC finite-sum saddle point problem

$$\min_{x \in \mathcal{X}} \max_{y \in \mathcal{Y}} \hat{F}_k(x, y) \triangleq \frac{1}{\bar{n}} \sum_{i=(k-1)\bar{n}+1}^{k\bar{n}} f(x, y; \xi_i) + \frac{\mu}{2}\|x\|^2 - \frac{\mu_k}{2}\|y - \tilde{y}_{k-1}\|^2,$$

    to obtain the output $(x_k, y_k)$ such that with probability $1 - \delta/8$,

$$\mu\|x_k - \hat{x}_k^*\|^2 + \mu_k\|y_k - \hat{y}_k^*\|^2 \leq \frac{L^2}{\mu\bar{n}^2},$$

    where $(\hat{x}_k^*, \hat{y}_k^*)$ is the saddle point of the regularized empirical function $\hat{F}_k(x, y)$.
5:     $\tilde{y}_k = y_k + \mathcal{N}(0, \sigma_k^2 I_{d_y})$ with $\sigma_k = (8L/(\bar{n}\varepsilon))\sqrt{2\log(5/\delta)/(\mu_k\mu)}$.
**Output:** $\tilde{y}_K$.

---

is the saddle point of $F(x, y)$, we know that $F(x^*, \hat{y}_1^*) \leq F(x^*, y^*)$, and thus

$$\max_{y \in \mathcal{Y}} \mathbb{E}[F(\tilde{x}_K, y)] \leq F(x^*, y^*) + 8LDK^2\left(\frac{1}{\sqrt{n}} + \frac{5\sqrt{d\log(5/\delta)}}{n\varepsilon}\right). \tag{28}$$

Here we give the privacy and utility guarantees of the primal solution $\tilde{x}_K$. The dual solution comes from a symmetric Algorithm 5. The same as the above analysis for Algorithm 4, we can show that Algorithm 5 is $(\varepsilon/2, \delta/2)$-DP and give the utility bound of its output $\tilde{y}_K$. Without causing confusion, we borrow notations from Algorithm 4 for simplicity. Since everything is very much similar by switching the role of the primal $x$ and the dual $y$, we will not repeat all the details.

First, each phase of Algorithm 5 is $(\varepsilon/2, \delta/2)$-DP since with probability at least $1 - \delta/4$, the sensitivity of $y_k$ is bounded by $4L/(\bar{n}\sqrt{\mu_k\mu})$. Similar to (24), the output $\tilde{y}_k$ for $1 \leq k \leq K$ satisfies that

$$\mathbb{E}\|\tilde{y}_k - \hat{y}_k^*\|^2 \leq \frac{257L^2}{\mu_k} \frac{d_y\log(5/\delta)}{\mu\bar{n}^2\varepsilon^2}. \tag{29}$$

Then by the parallel composition of differential privacy, Algorithm 5 is $(\varepsilon/2, \delta/2)$-DP. For the utility bound of the output $\tilde{y}_K$, we have the following error decomposition that mirrors (25):

$$\mathbb{E}[F(\hat{x}_1^*, y^*)] - \min_{x \in \mathcal{X}} \mathbb{E}[F(x, \tilde{y}_K)] = \sum_{k=1}^K \mathbb{E}\Big[F(\hat{x}_k^*, \hat{y}_{k-1}^*) - F(\hat{x}_{k+1}^*, \hat{y}_k^*)\Big]$$
$$+ \mathbb{E}[F(\hat{x}_{K+1}^*, \hat{y}_K^*) - F(\hat{x}_{K+1}^*, \tilde{y}_K)], \tag{30}$$

where $(\hat{x}_k^*, \hat{y}_k^*)$ is the saddle point of the regularized empirical function $\hat{F}_k(x, y)$ for $1 \leq k \leq K$, and we let $\hat{y}_0^* = y^*$ for the saddle point of $F(x, y)$ and $\hat{x}_{K+1}^* \in \arg\min_{x \in \mathcal{X}} \mathbb{E}[F(x, \tilde{y}_K)]$ to simplify the analysis. The first term in the RHS of (30) can be bounded by a similar result as Lemma B.5. By Corollary B.3 and Lemma B.4, setting $x = \hat{x}_{k+1}^*$ and $y = \hat{y}_{k-1}^*$, we obtain that

$$\mathbb{E}[F(\hat{x}_k^*, \hat{y}_{k-1}^*) - F(\hat{x}_{k+1}^*, \hat{y}_k^*)] \leq \frac{\mu_k}{2}\mathbb{E}\|\hat{y}_{k-1}^* - \tilde{y}_{k-1}\|^2 + \frac{8L^2}{\mu\bar{n}} + \frac{\mu}{2}\mathbb{E}\big[\|\hat{x}_{k+1}^*\|^2 - \|\hat{x}_k^*\|^2\big].$$

The second term in the RHS of (30) can be bounded by Lipschitzness of $F(x, y)$,

$$\mathbb{E}[F(\hat{x}_{K+1}^*, \hat{y}_K^*) - F(\hat{x}_{K+1}^*, \tilde{y}_K)] \leq 17LD \cdot \frac{\sqrt{d\log(5/\delta)}}{2\bar{n}\varepsilon},$$

which is the same as (26) using the guarantee of $\tilde{y}_K$ in (29). Finally plugging the above two bounds back into (30), by the same reason as (27), we get that

$$\mathbb{E}[F(\hat{x}_1^*, y^*)] - \min_{x \in \mathcal{X}} \mathbb{E}[F(x, \tilde{y}_K)] \leq \sum_{k=2}^{K} \frac{257\mu_k}{2\mu_{k-1}} \frac{L^2 d_y \log(5/\delta)}{\mu \bar{n}^2 \varepsilon^2} + \sum_{k=1}^{K} \frac{8L^2}{\mu \bar{n}}$$

$$+ \frac{\mu_1}{2} \|y^* - y_0\|^2 + \frac{\mu}{2} \|\hat{x}_{K+1}^*\|^2 + 17LD \cdot \frac{\sqrt{d \log(5/\delta)}}{2\bar{n}\varepsilon}$$

$$\leq 8LDK^2 \left( \frac{1}{\sqrt{n}} + \frac{5\sqrt{d \log(5/\delta)}}{n\varepsilon} \right),$$

where we use the guarantees of $\tilde{y}_k$ in (29) and the assumption that $\|y^*\|^2 \leq D^2$ and $\|\hat{x}_{K+1}^*\|^2 \leq D^2$ when the initialization is $y_0 = 0$. Finally since $(x^*, y^*)$ is the saddle point of $F(x, y)$, we know that $F(\hat{x}_1^*, y^*) \geq F(x^*, y^*)$, and thus

$$- \min_{x \in \mathcal{X}} \mathbb{E}[F(x, \tilde{y}_K)] \leq -F(x^*, y^*) + 8LDK^2 \left( \frac{1}{\sqrt{n}} + \frac{5\sqrt{d \log(5/\delta)}}{n\varepsilon} \right). \quad (31)$$

By basic composition in Lemma 2.1, the composition $(\tilde{x}_K, \tilde{y}_K)$ of Algorithm 4 and 5 is $(\varepsilon, \delta)$-DP. The proof is thus complete summing up (28) and (31). Note that we only require that $(x^*, y^*)$ and $(\hat{x}_{K+1}^*, \hat{y}_{K+1}^*)$ have bounded norms, which is slightly weaker than assuming bounded domains. □

**Near-optimality:** The lower-bound [6, 7] $\Omega(1/\sqrt{n} + \sqrt{d \log(1/\delta)}/(n\varepsilon))$ of DP-SCO problems can be also regarded as a lower-bound of DP-SMO problems (see Boob and Guzmán [11] for more details). Thus we can achieve optimal rates up to logarithmic factors.

**Gradient Complexity:** To compute the gradient complexity of our Algorithm 4 and 5, we first review some existing algorithms for smooth SC-SC finite-sum saddle point problems. The general finite-sum minimax optimization problem can be formulated as

$$\min_{x \in \mathcal{X}} \max_{y \in \mathcal{Y}} \hat{F}_S(x, y) = \frac{1}{n} \sum_{i=1}^{n} f(x, y; \xi_i),$$

where $f(x, y; \xi)$ is $\mu_x$-strongly convex in $x$, $\mu_y$-strongly concave in $y$ and $\ell$-smooth. We are interested in the $\gamma$-approximate saddle point $(\tilde{x}, \tilde{y}) \in \mathcal{X} \times \mathcal{Y}$ such that

$$\mathbb{E}\left[ \max_{y \in \mathcal{Y}} \hat{F}_S(\tilde{x}, y) - \min_{x \in \mathcal{X}} \hat{F}_S(x, \tilde{y}) \right] \leq \gamma,$$

where the expectation is taken w.r.t. the randomness in the algorithm. GDA and Extragradient [43] use the full-batch gradient at each iteration and consider a deterministic problem. Since the iteration complexity is $\mathcal{O}(\kappa^2 \log(1/\gamma))$ and $\mathcal{O}(\kappa \log(1/\gamma))$ respectively, the total gradient complexity is $\mathcal{O}(n\kappa^2 \log(1/\gamma))$ for GDA and $\mathcal{O}(n\kappa \log(1/\gamma))$ for Extragradient to achieve a $\gamma$-approximate saddle point, where $\kappa = \ell/\min\{\mu_x, \mu_y\}$. These two algorithms do not leverage the finite-sum structure and obtain sub-optimal convergence rates. Palaniappan and Bach [39] first introduced the use of variance reduction methods into finite-sum minimax optimization problems. The gradient complexity is improved to $\mathcal{O}((n + \kappa^2) \log(1/\gamma))$ by SVRG/SAGA and $\mathcal{O}((n + \sqrt{n}\kappa) \log(1/\gamma))$ by the accelerated SVRG/SAGA. The state-of-the-art complexity $\mathcal{O}((n + \sqrt{n\kappa_x \kappa_y} + n^{3/4}\sqrt{\kappa}) \log(1/\gamma))$ that also matches with the lower-bound is provided in Yang et al. [48] and Luo et al. [34], where $\kappa_x = \ell/\mu_x$ and $\kappa_y = \ell/\mu_y$. Their algorithms are based on a catalyst framework.

We then analyze the complexity of Algorithm 4 and 5. Since two algorithms have the same gradient complexity, we only do the computations for Algorithm 4. If we regard the regularized empirical problem in Algorithm 4 as a general $(\ell + \mu_k)$-smooth, $\mu_k$-strongly convex and $\mu$-strongly concave finite-sum minimax problem, linear time-complexity cannot be achieved since the condition number $(\ell + \mu_k)/\mu \leq \mathcal{O}(\sqrt{n}\ell D/L) + 2^k$ can be as large as $\mathcal{O}(n)$. By Remark 3, the complexity of each phase in Algorithm 4 is $\mathcal{O}(T(\bar{n}, \kappa_x, \kappa_y) \log(1/\gamma))$ where $\kappa_x = (\ell + \mu_k)/\mu_k$, $\kappa_y = (\ell + \mu_k)/\mu$ and

Table 2: Comparisons of the gradient complexity of Algorithm 3 for smooth SC-SC DP-SMO and Algorithm 4 and 5 for smooth convex-concave DP-SMO when equipping with different non-private minimax optimization algorithms. Here $\ell$ is the smoothness parameter of $f(x,y;\xi)$, $\kappa_x = \ell/\mu_x$, $\kappa_y = \ell/\mu_y$, $\kappa = \max\{\kappa_x, \kappa_y\}$ are condition numbers, and $\tilde{\mathcal{O}}$ hides logarithmic factors in $n/\delta$.

| Algorithm | SC-SC | Convex-Concave |
|---|---|---|
| Extragradient [43] | $\tilde{\mathcal{O}}(n\kappa)$ | $\tilde{\mathcal{O}}(n^{3/2}\ell D/L + n^2)$ |
| SVRG/SAGA [39] | $\tilde{\mathcal{O}}(n + \kappa^2)$ | $\tilde{\mathcal{O}}(n + n(\ell D/L)^2)$ |
| Acc-SVRG/SAGA [39] | $\tilde{\mathcal{O}}(n + \sqrt{n}\kappa)$ | $\tilde{\mathcal{O}}(n + n\ell D/L)$ |
| Jin et al. [25] | $\tilde{\mathcal{O}}(n + \sqrt{n}\kappa)$ | $\tilde{\mathcal{O}}(n + n\ell D/L)$ |
| AL-SVRE [34] | $\tilde{\mathcal{O}}(n + \sqrt{n\kappa_x\kappa_y} + n^{3/4}\sqrt{\kappa})$ | $\tilde{\mathcal{O}}(n\ell D/L + n^{5/4})$ |
| Catalyst-Acc-SVRG [48] | $\tilde{\mathcal{O}}(n + \sqrt{n\kappa_x\kappa_y} + n^{3/4}\sqrt{\kappa})$ | $\tilde{\mathcal{O}}(n\ell D/L + n^{5/4})$ |

$\gamma = \delta L^2/(16\mu\bar{n}^2)$ since $\mu \le \mu_k$. As a result, when using the state-of-the-art algorithms AL-SVRE [34] and Catalyst-Acc-SVRG [48], the total gradient complexity is

$$\sum_{k=1}^{K} \mathcal{O}\left(\left(\bar{n} + \sqrt{\frac{\bar{n}(\ell + \mu_k)^2}{\mu_k\mu}} + \bar{n}^{3/4}\sqrt{\frac{\ell + \mu_k}{\mu}}\right)\log(1/\gamma)\right)$$

$$\le \sum_{k=1}^{K} \mathcal{O}\left(\left(\bar{n} + \bar{n}^{3/4}\sqrt{\frac{\ell}{\mu}} + \frac{\sqrt{\bar{n}}\ell}{\mu}\frac{1}{\sqrt{2^k}} + (\sqrt{\bar{n}} + \bar{n}^{3/4})\sqrt{2^k}\right)\log(n/\delta)\right)$$

$$\le \mathcal{O}((n + n\sqrt{\ell D/L} + n\ell D/L + n^{5/4})\log(n/\delta)),$$

since $1/\mu \le \mathcal{O}(\sqrt{n}D/L)$ and $\sum_{k=1}^{K}\sqrt{2^k} = \mathcal{O}(\sqrt{n})$. The complexity of Extragradient [43] can be computed similarly. We find that the super-linear complexity comes from the potentially large smoothness parameter corresponding to the regularizer $(\mu_k/2)\|x - \tilde{x}_{k-1}\|^2$. Next we show this can be avoided by algorithms with fine-grained analyses and thus near-linear time-complexity is possible.

Jin et al. [25] provided sharper rates when the problem is separable. The problem they considered is

$$\min_{x\in\mathcal{X}} \max_{y\in\mathcal{Y}} \hat{F}_1(x,y) \triangleq \frac{1}{n}\sum_{i=1}^{n} f(x,y;\xi_i) + g(x) - h(y),$$

where $f(x,y;\xi)$ is $\ell$-smooth and convex-concave, $g(x)$ is $\ell_g$-smooth, $h(y)$ is $\ell_h$-smooth and the overall function $\hat{F}_1(x,y)$ is $\mu_x$-strongly convex $\mu_y$-strongly concave. Their algorithms achieve a $\gamma$-approximate saddle point of $\hat{F}_1(x,y)$ with complexity

$$\tilde{\mathcal{O}}\left(\left(n + \sqrt{n}\left(\sqrt{\frac{\ell_g}{\mu_x}} + \sqrt{\frac{\ell_h}{\mu_y}} + \frac{\ell}{\mu_x} + \frac{\ell}{\sqrt{\mu_x\mu_y}} + \frac{\ell}{\mu_y}\right)\right)\log(1/\gamma)\right),$$

where $\tilde{\mathcal{O}}$ hides additional logarithmic terms. Our regularized problems satisfy the above special structure with $\ell_g = \mu_x = \mu_k$ and $\ell_h = \mu_y = \mu$. This fine-grained analysis is especially suitable for us since the potentially large smoothness parameter $\mu_k$ does not affect the complexity result. The total complexity of Algorithm 4 using results in Jin et al. [25] is thus

$$\sum_{k=1}^{K} \tilde{\mathcal{O}}\left(\left(\bar{n} + \sqrt{\bar{n}}\left(\frac{\ell}{\mu_k} + \frac{\ell}{\sqrt{\mu_k\mu}} + \frac{\ell}{\mu}\right)\right)\log(1/\gamma)\right) \le \tilde{\mathcal{O}}\left(\left(n + \frac{\sqrt{n}\ell}{\mu}\right)\log(n/\delta)\right)$$

$$\le \tilde{\mathcal{O}}((n + n\ell D/L)\log(n/\delta)),$$

which is linear in the number of samples up to logarithmic factors.

The above results suggest that the smoothness parameter of the regularization term in our problems does not affect the gradient complexity. Actually Palaniappan and Bach [39] studied the case

$$
\min_{x \in \mathcal{X}} \max_{y \in \mathcal{Y}} \ \hat{F}_2(x, y) \triangleq \frac{1}{n} \sum_{i=1}^{n} f(x, y; \xi_i) + g(x, y),
$$

where $f(x, y; \xi)$ is convex-concave and $\ell$-smooth, and $g(x, y)$ is $\mu_x$-strongly convex, $\mu_y$-strongly concave and possibly *nonsmooth*. Moreover, the nonsmooth part $g(x, y)$ is required to be "simple" in the sense that the proximal operator

$$
\mathrm{prox}_g^{\lambda}(x', y') \triangleq \arg\min_{x \in \mathcal{X}} \max_{y \in \mathcal{Y}} \Big\{ \lambda g(x, y) + \frac{\mu_x}{2} \|x - x'\|^2 - \frac{\mu_y}{2} \|y - y'\|^2 \Big\},
$$

is easy to compute for any $(x', y') \in \mathcal{X} \times \mathcal{Y}$. The complexity of their algorithms SVRG/SAGA and Acc-SVRG/SAGA only depends on the smoothness parameter of $f(x, y; \xi)$ and SC-SC parameters $\mu_x$, $\mu_y$, i.e., $\ell/\min\{\mu_x, \mu_y\}$. The algorithms are first designed for matrix games, but also work for general functions (see extensions to monotone operators in their supplementary material [39]). Our regularized problems satisfy the special structure since the quadratic regularization term is prox-friendly. The resulting complexity is

$$
\sum_{k=1}^{K} \mathcal{O}\bigg( \Big( \bar{n} + \frac{\sqrt{\bar{n}}\ell}{\mu} \Big) \log(1/\gamma) \bigg) = \mathcal{O}((n + n\ell D/L) \log(n/\delta)),
$$

for Acc-SVRG/SAGA and $\mathcal{O}((n + n(\ell D/L)^2) \log(n/\delta))$ for SVRG/SAGA by a similar computation. Table 2 summarizes the complexity of our private algorithms for both smooth SC-SC and convex-concave cases when equipping with different non-private minimax optimization algorithms. We provide several instances where near-linear time-complexity is possible. It is worth mentioning that AL-SVRE [34] and Catalyst-Acc-SVRG [48] can achieve the near-linear time-complexity for convex-concave problems when exploiting the special structure of the regularized problems. We leave this for future work. Minimax optimization with finite-sum structure is a well-studied research topic, and we only include a small set of representative algorithms here. We refer interested readers to Yang et al. [48], Luo et al. [34] and references therein for more discussions.

## B.4 Near-Linear Time Algorithms for Smooth Convex–Strongly-Concave Functions

Algorithm 4 can be extended to the convex–strongly-concave (C-SC) case, and Algorithm 5 can be extended to the strongly-convex–concave (SC-C) case. In this section, we provide a brief discussion for the setting that $f(x, y; \xi)$ is convex in $x$ and $\mu_y$-strongly concave in $y$. The SC-C setting is similar and we omit it here. When $\mu_y \leq \mathcal{O}(1/\sqrt{n})$, the strong-concavity does not help too much and we can directly regard it as a convex-concave problem and apply Algorithm 4 and 5. However, there is no need to add the regularization term for $y$ in Algorithm 4 when $\mu_y \geq \mathcal{O}(1/\sqrt{n})$. The resulting variant for smooth C-SC DP-SMO iteratively solves

$$
\min_{x \in \mathcal{X}} \max_{y \in \mathcal{Y}} \left\{ \frac{1}{\bar{n}} \sum_{i=(k-1)\bar{n}+1}^{k\bar{n}} f(x, y; \xi_i) + \frac{\mu_k}{2} \|x - \tilde{x}_{k-1}\|^2 \right\},
$$

at each phase $k$ to obtain the output $(x_k, y_k)$ such that with probability $1 - \delta/4$,

$$
\mu_k \|x_k - \hat{x}_k^*\|^2 + \mu_y \|y_k - \hat{y}_k^*\|^2 \leq \frac{L^2}{\bar{n}^2 \min\{\mu_k, \mu_y\}},
$$

where $(\hat{x}_k^*, \hat{y}_k^*)$ is the saddle point of the regularized empirical function. Then the perturbed output $\tilde{x}_k$ is obtained by adding Gaussian noise $\mathcal{N}(0, \sigma_k^2 I_{d_x})$ to the primal solution $x_k$ with variance $\sigma_k = (4L/(\bar{n}\varepsilon))\sqrt{2\log(2.5/\delta)/(\mu_k \min\{\mu_k, \mu_y\})}$. As a direct corollary of Theorem 4.5, the final output $\tilde{x}_K$ satisfies the following guarantees.

**Corollary B.6.** *Let Assumption 4.1 hold. Assume that $f(x, \cdot; \xi)$ is $\mu_y$-strongly concave on $\mathcal{Y}$ for any $x \in \mathcal{X}$ with $\mu_y \geq L/(D\sqrt{n})$. Suppose $\max\{\|x\|, \|y\|\} \leq D$ for all $x \in \mathcal{X}$ and $y \in \mathcal{Y}$. Then the*

*variant of Algorithm 4 discussed above is $(\varepsilon, \delta)$-DP and its output $\tilde{x}_K$ satisfies that*

$$\max_{y \in \mathcal{Y}} \mathbb{E}[F(\tilde{x}_K, y)] - F(x^*, y^*) \leq 6LD \log(n) \left( \frac{1}{\sqrt{n}} + \frac{4\sqrt{d_x \log(2.5/\delta)}}{n\varepsilon} \right)$$
$$+ L^2 \log^2(n) \left( \frac{8}{\mu_y n} + \frac{65 d_x \log(2.5n/\delta)}{\mu_y n^2 \varepsilon^2} \right),$$

*when setting $\mu = 3(L/D) \max \left\{ 1/\sqrt{n}, 4\log(n)\sqrt{d_x \log(2.5/\delta)}/(n\varepsilon) \right\}$.*

The proof directly follows from (27) when replacing $\mu$ by $\min\{\mu_k, \mu_y\}$ for every $\mu$ appearing in the denominator and the fact that $1/\min\{\mu_k, \mu_y\} \leq 1/\mu_k + 1/\mu_y$. The requirement that $\mu_y \geq \mathcal{O}(1/\sqrt{n})$ is used to control the DP noise such that (26) holds (noticing that $1/(\mu_K \min\{\mu_K, \mu_y\}) \leq \mathcal{O}(1)$ when $\mu_y$ is not too small). When $f(x, y; \xi)$ is C-SC, the primal function $\Phi(x) = \max_{y \in \mathcal{Y}} \mathbb{E}[F(x, y)]$ is smooth and convex. Since $F(x^*, y^*) = \Phi(x^*)$, the above guarantees imply that

$$\mathbb{E}[\Phi(\tilde{x}_K) - \Phi(x^*)] \leq 6LD \log(n) \left( \frac{1}{\sqrt{n}} + \frac{4\sqrt{d_x \log(2.5/\delta)}}{n\varepsilon} \right)$$
$$+ L^2 \log^2(n) \left( \frac{8}{\mu_y n} + \frac{65 d_x \log(2.5n/\delta)}{\mu_y n^2 \varepsilon^2} \right),$$

which is the optimal rate, up to logarithmic factors, for the smooth convex DP-SCO problem $\min_{x \in \mathcal{X}} \Phi(x)$. Then we analyze its gradient complexity by similar computations as the above section. The total complexity is $\tilde{\mathcal{O}}(n + n(\ell D/L)^2 + \kappa_y^2)$ for SVRG/SAGA [39], $\tilde{\mathcal{O}}(n + n\ell D/L + \sqrt{n}\kappa_y)$ for Acc-SVRG/SAGA [39] and $\tilde{\mathcal{O}}(n + n\ell D/L + \sqrt{n}\kappa_y)$ using Jin et al. [25], where $\tilde{\mathcal{O}}$ hides additional logarithmic terms and $\kappa_y = \ell/\mu_y$. Not surprisingly, near-linear time-complexity is achievable.