# OpenReview forum: "Bring Your Own Algorithm for Optimal Differentially Private Stochastic Minimax Optimization"
_NeurIPS.cc/2022/Conference — NeurIPS 2022 Accept_

### Official Review · Reviewer_Jh8C · 2022-07-08

**Rating:** 7
**Confidence:** 4
**Soundness:** 4 excellent
**Presentation:** 4 excellent
**Contribution:** 3 good

**Summary:**

The paper is one more link in the ever growing chain of results on DP ERM-based optimization algorithms, which dates all the way back to CMT08 and BST14. In this paper, the authors up the ante on two main axes, as follows. First, rather than looking at classic minimization problem, they deal with a min-max problem (a setting that does arise in practice, such as adversarial GANs), see Eqs. (1)-(2). Second, rather than giving their own algo. for such an optimization, the authors assume some off-the-shelf algorithm for which they can apply output perturbation since - by assumption - the optimization function is smooth and stongly-convex/strong-concave, a setting where an approximated loss must imply approximated models. Lastly, the authors achieve near-optimal loss in near-linear time ("near"=up to poly-log factors).


**Questions:**

The runtime complexity's dependency on \epsilon for DP-SGDA is cited correctly?

**Limitations:**

I wish they had a line saying "while having the loss-function strongly-convex / strongly-concave and smooth, we make rather a strong assumption, but this is ok as this is the first paper to ..."

**Strengths And Weaknesses:**

Ultimately this is an interesting paper that makes solid progress on one of the more canonical and current problems (or venues) studied in the DP-ML literature. While the novelty factor isn't that great (the assumption of a loss-function which is strongly-convex strong-concave and smooth does imply the result) I still believe it is a problem worth mentioning in NeurIPS, in particular as it may "lead the way" for min-max DP-optimization algorithms that require less on the loss-function.

Strengths: well written, solid progress on a very relevant problem.
Weaknesses: rather strong assumption on the loss-function

---

> ### Author Response · Authors · 2022-08-02
> **Response to Reviewer Jh8C**
>
> We thank the reviewer for acknowledging the value of this work.
>
> 1. **Strong assumptions on the function.**
>
> > The novelty factor isn't that great (the assumption of a loss-function which is strongly-convex strong-concave and smooth does imply the result).
>
> We first thank the reviewer for identifying our contributions on DP-SMO. We kindly remind the reviewer about our contributions on smooth **convex-concave** minimax optimization in Section 4.2, which puts weaker assumptions on the loss functions compared to the strongly-convex strongly-concave case. For this setting, the results are not straightforward but we can achieve the **first** near-linear time algorithm with near-optimal utility bound, improving upon previous works (see Table 1 for a comparison). This also implies that near-linear time algorithms exist for **nonsmooth** DP-SCO when the problem allows for some smooth convex-concave minimax reformulations. The task of designing linear-time algorithms for nonsmooth DP-SCO with optimal rates has been a **long-standing** open question in the DP community [7, 21, 4, 29]. We believe our findings make an initial step towards this goal. For general settings beyond convex-concave assumptions (e.g., nonconvex--strongly-concave, nonconvex-concave), we leave them for future works as even nonconvex DP minimization problems are less understood.
>
>
> 2. **Runtime dependence on $\varepsilon$ for DP-SGDA.**
>
> According to Theorem 2 in Yang et al. [47], for smooth convex-concave functions, DP-SGDA runs in $T=n$ iterations. However, by Remark 1 in [47], the batch size has to be $m=\max(1, n/\sqrt{\varepsilon T})$ for the privacy guarantee. Therefore, the total gradient complexity is $mT=\max(n, n^{3/2}\sqrt{\varepsilon})\geq n^{3/2}\sqrt{\varepsilon}$.

---

> > ### Comment · Reviewer_Jh8C · 2022-08-07
> > **No further questions**
> >
> > I have no further questions for the authors.

---

### Official Review · Reviewer_s8NK · 2022-07-08

**Rating:** 5
**Confidence:** 1
**Soundness:** 3 good
**Presentation:** 2 fair
**Contribution:** 2 fair

**Summary:**

The paper provides a general framework for solving smooth DP-SMO and smooth DP-SCO problems by utilizing the (phased) output perturbation mechanism, so there is no need for modifying the algorithms themself to have a DP result.

**Questions:**

1. Shouldn't the input of the proposed algorithms (i.e., Algorithm 1) include the parameter $\mu$?

2. How could one decide $L$ in Algorithm 1 in practice?



**Limitations:**

Yes.

**Strengths And Weaknesses:**

Strengths: theoretically sounded and working for a wide range of objective functions.

Weaknesses: 1. Lack of numerical results. The reviewer is curious about how to apply it to some popular algorithms and their performance compared with existing DP algorithms.
2. The presentation of this paper is hard to follow for the reviewer.

---

> ### Author Response · Authors · 2022-08-02
> **Response to Reviewer s8NK**
>
> We thank the reviewer for the questions.
>
> 1. **Lack of numerical results.**
>
> > The reviewer is curious about how to apply it to some popular algorithms and their performance compared with existing DP algorithms.
>
> The studying of DP optimization algorithms with theoretical guarantees is an active research area [5, 6, 7, 21, 4, 29, 8, 10], just to name a few. The purpose of this series of **purely theoretical** work is to find out the fundamental limits of DP optimization, by pushing towards both the lower and upper-bounds. Our paper follows this line of work and answers the question about the optimal utility bound and gradient complexity for smooth convex-concave DP-SMO, resolving an open question in the DP community. Building upon this, we do not think numerical experiments will bring additional value to justify our contribution. However, we are happy to provide some if the reviewer strongly insists.
>
>
> 2. **The presentation is hard to follow.**
>
> We thank the reviewer for this feedback. It would be great if the reviewer could provide more detailed suggestions. We are happy to have more discussions on how to improve the presentation of our paper.
>
>
> 3. **Parameters $\mu$ and $L$.**
>
> > Shouldn't the input of the proposed algorithms (i.e., Algorithm 1) include the parameter $\mu$? How could one decide $L$ in Algorithm 1 in practice?
>
> We agree that the problem-specific parameter $\mu$ should also appear in the input of the algorithm. The knowledge of Lipschitz constant $L$ is essential to the design of many DP algorithms, including DP-SGD [6, 7, 21], phased-ERM [21, 4, 29], DP-SGDA [47], NSEG [10], and NISPP [10]. In practice, any estimate of the upper bound of Lipschitz constant $L$ can be used in the algorithm, e.g., see the following references. We will include more discussions on this issue in the revision.
>
> * Wood, G. R., and B. P. Zhang. "Estimation of the Lipschitz constant of a function." Journal of Global Optimization, 1996.
>
> * Fazlyab, Mahyar, et al. "Efficient and accurate estimation of lipschitz constants for deep neural networks." Advances in Neural Information Processing Systems, 2019.

---

> > ### Comment · Reviewer_s8NK · 2022-08-05
> > **My questions are fully addressed.**
> >
> > Thanks for the reply.  My questions are fully addressed, and the reviewer is willing to accept the paper.

---

### Official Review · Reviewer_XKZT · 2022-07-11

**Rating:** 5
**Confidence:** 4
**Soundness:** 3 good
**Presentation:** 3 good
**Contribution:** 3 good

**Summary:**

The paper studies DP stochastic convex optimization (SCO) and stochastic minmax optimization (SMO). They provide near-linear time algorithms under (strong) convexity assumptions that attain optimal or near-optimal rates.

**Questions:**

I think most of my questions were asked in the above. Two more below:

-Could you elaborate on the reason for the asymmetry in the x and y regularizers in Algorithm 4?

-You claim in Remark 2 that smoothness is not necessary for the utility bound to hold. Can you explain why? Also, what gradient complexity would be achievable without smoothness?

**Strengths And Weaknesses:**

Strengths:

-The phased output perturbation idea is nice and simple, but apparently also effective.

-Apparently improved (faster) algorithms for convex-concave and SC-SC min-max are the main contributions.

Weaknesses (and borderline strengths/contributions):

-Missing related work:

Theorem 3.3 (excess risk bounds with black box "bring your own algorithm" output perturbation for strongly convex loss) seems to have already been provided in prior work: see Corollary 3.4 and Corollary 4.2 of https://arxiv.org/pdf/2102.04704.pdf. The idea of "bring your own algorithm" output perturbation is also a theme in this work and should be credited.

-Theorem 3.4 is a somewhere between a good contribution and a marginal contribution: it is essentially optimal excess risk and near-linear time: existing works (e.g. FKT20) give optimal excess risk in linear time with some mild smoothness restrictions that this excess risk bound removes, but smoothness is still required here. Furthermore, a restriction on the smoothness parameter seems to be required here in order to be linear time. Also, optimal algorithms for non-smooth losses require a bit more computation (e.g. $n^{11/8}$ is the smallest gradient complexity I believe) vs. Theorem 3.4 is $n + \sqrt{n} \ell$. On a related note, the units in the gradient complexity expression are not matching, as $\ell$ is not unit-free. What is the precise unit-free gradient complexity (e.g. with SVRG)?

-**Discussion of optimality of the min-max bounds seems to be lacking. Lower bound for convex-concave was provided in Boob & Guzman, but I'm not sure its obvious that the the SC-SC lower bound follows immediately from SC SCO lower bound? It would be great to complete Table 1 with lower bounds.

-Some imprecision/slight overclaiming of "optimal"  and "linear time" in places where this should be qualified with "near" (e.g. due to dependence on condition number). Also, "smooth" should be included  for emphasis whenever the problem class is smooth--e.g. Section 3.2 heading should be "near-linear time algorithms for smooth convex functions" since I don't think you are claiming near-linear time algorithms for general (non-smooth) convex functions.

---

> ### Author Response · Authors · 2022-08-02
> **Response 2 to Reviewer XKZT**
>
> 3. **Utility bound without smoothness.**
>
> > You claim in Remark 2 that smoothness is not necessary for the utility bound to hold. Can you explain why? Also, what gradient complexity would be achievable without smoothness?
>
> Since the stability and generalization results (Lemma A.2, A.3) do not require the smoothness assumption, the utility bound in Theorem 3.4 holds as long as the output of $\mathcal{A}$ is close enough to the empirical solution (line 4 of Algorithm 2). Smoothness is only used to make sure we can efficiently compute such output (with linear convergence). Without smoothness, our guarantee is the same as phased-ERM [21] for nonsmooth DP-SCO, where the optimal rate requires $n^2$ gradient complexity.
>
>
> 4. **Lower-bounds for DP-SMO.**
>
> >  Discussion of optimality of the min-max bounds seems to be lacking. I'm not sure its obvious that the the SC-SC lower bound follows immediately from SC SCO lower bound?
>
> The discussions of lower-bounds and near-optimality of our results were provided in the appendix (see Section B.2, line 701-707,  and Section B.3, line 800-804 for SC-SC and C-C cases, respectively). We will move them to the main text and Table 1 in the updated version. As a special case by restricting the domain of $y$ to a singleton, the minimax problem is equivalent to a minimization problem on $x$. Thus, the lower-bound $\Omega(1/(\mu\_x n)+d\_x\log(1/\delta)/(\mu\_x n^2\varepsilon^2))$ of SC DP-SCO **yields a trivial lower bound**  for SC-SC DP-SMO. This lower bound already implies the (near)-optimality of our algorithm in terms of the dependence on  $n, \varepsilon, \delta$. However, this lower bound is of course by no means tight, as it hardly captures the minimax structure and the dependence on $\mu\_y, d\_y$. It remains interesting to investigate a tight lower bound for DP-SMO and an algorithm with also optimal dependence on other parameters such as $\mu\_x,\mu\_y, d\_x, d\_y$. However, this is beyond the scope of the current work (as some of these questions are not even fully addressed in the non-DP setting).
>
> 5. **Asymmetry in the regularizers in Algorithm 4.**
>
> Note that Algorithm 4 is only used to find a primal solution. The regularization in $x$ and $y$ plays different roles.
>
> * The $\ell_2$ norm regularization in $y$ is to ensure strong-concavity and thereby the smoothness of the primal function, which is similar to Nesterov’s smoothing [36]. The small regularization parameter is required to balance the smoothness and the approximation error, and an arbitrary proximal center (we use origin $0$) suffices for the purpose.
>
> * The "proximal regularization" in $x$ is similar to Algorithm 2 for **smooth** convex functions, where the increasing regularization parameter $\mu\_k$ and the choice of $\tilde x\_{k-1}$ as the proximal center are necessary for the near-optimal utility bound of the final output.
>
> To obtain a dual solution,  we use an analogous algorithm (Algorithm 5 in Appendix B.3), where the $x$ and $y$ regularizers are exchanged. Here, the $\ell_2$-norm regularization in $x$ ensures strong-convexity and the "proximal regularization"  in $y$ is essential for the near-optimal guarantees of the output.
>
>
> 6. **Some imprecision/slight overclaiming.**
>
> > "Optimal" and "linear time" should be qualified with "near" (e.g. due to dependence on condition number). "Smooth" should be included for emphasis whenever the problem class is smooth.
>
> We really appreciate the reviewer for this comment. We agree that the optimality of the bound should also take into account the condition number of the problem. We will be careful about the phrases and make sure everything is precise in the revision.

---

> > ### Comment · Reviewer_XKZT · 2022-08-07
> > **Response**
> >
> > Thank you for your response. You have addressed most of my questions and concerns.
> >
> > However, I'm still not convinced about the SC-SC lower bound following immediately from the SC minimization lower bound. To establish the desired SC-SC lower bound, you need to show there is a SC-SC function with large error. The hard instance for SC minimization $f(x; z)$ can be regarded as a min-max function $f(x, y; z): \mathcal{X} \times \mathcal{Y} \times \mathcal{Z} \to \mathbb{R}$ for $\mathcal{Y} = \{y\}$, but this min-max function is not strongly concave in $y$: it is a constant function. Can you please clarify?

---

> > > ### Author Response · Authors · 2022-08-08
> > > **Some Clarification**
> > >
> > > Many thanks for the follow-up question. We provide here two examples such that SC lower-bound can be regarded as a trivial lower-bound of SC-SC functions.
> > >
> > > 1. Given an SC-SC function $F(x,y)$ defined on $\mathcal{X}\times\mathcal{Y}$, we restrict the **domain** $\mathcal{Y}$ to be a **singleton**, i.e., $\mathcal{Y}=\\{y\_0\\}$. The function is still strongly-concave in $y$ on the domain, but the SC-SC minimax problem $\min\_{x\in\mathcal{X}}\max\_{y\in\mathcal{Y}} F(x,y)$ is equivalent to the SC minimization problem $\min\_{x\in\mathcal{X}} F(x,y_0)$.
> > >
> > > 2. Given a "hard" example $f(x)$ for SC minimization problem, we construct an **uncoupled** SC-SC function as $F(x,y)=f(x)-h(y)$ for a strongly-convex function $h(y)$, e.g., $h(y)=(\mu\_y/2)\Vert y\Vert^2$. The resulting SC-SC minimax problem $\min\_{x\in\mathcal{X}}\max\_{y\in\mathcal{Y}} F(x,y)$ is equivalent to the SC minimization problem $\min\_{x\in\mathcal{X}} \\{f(x) - h^* \\}$ where $h^*$ is the minimum of $h(y)$ on the domain $\mathcal{Y}$.
> > >
> > > However, the above two examples hardly capture the minimax structure and thus the SC lower-bound only trivially holds. We are happy to have more discussions if the reviewer has further concerns.

---

> ### Author Response · Authors · 2022-08-02
> **Response 1 to Reviewer XKZT**
>
> We thank the reviewer for valuable suggestions. We will provide a revision that clarifies the problems.
>
> 1. **Missing related work.**
>
> > The idea of "bring your own algorithm" output perturbation is also a theme in arXiv:2102.04704 for strongly convex loss and should be credited.
>
> We thank the reviewer for pointing out this closely related paper. We have cited it in the revision, changed our statements accordingly, and added discussions on the comparison to our work. The idea of using the stability of ERM for privacy and generalization guarantees (first introduced in phased-ERM [21] for nonsmooth losses) for **smooth** DP-SCO is similar, but our work is significantly different in the following aspects.
>
> * **Privacy:** The base algorithms in their work only gives **in-expectation** guarantees, which brings a critical challenge to the design of private mechanisms (this issue was not properly addressed in their paper). However, the base algorithms $\mathcal{A}$ in our paper have **high-probability** guarantees; as a result, a slight modification to the Gaussian mechanism is enough to provide privacy (see Definition 2).
>
> * **DP-SCO:** Their work restricts to the output perturbation and only obtains near-linear time near-optimal algorithms for smooth strongly-convex functions. For smooth convex losses, they achieve a **sub-optimal** rate $\mathcal{O}(1/\sqrt{n} + (\sqrt{d\log(1/\delta)}/(n\varepsilon))^{2/3})$ in near-linear time. However, we use a **phased** framework in Algorithm 2 and achieve near-optimal rate $\mathcal{O}(1/\sqrt{n} + \sqrt{d\log(1/\delta)}/(n\varepsilon))$ in near-linear time.
>
> * **DP-SMO:** Their work also discusses smooth strongly-convex (strongly-)concave DP-SMO through reduction to the primal problem $\min\Phi(x)=\min\max\_y F(x,y)$. They simply use the analyses of strongly-convex DP-SCO to derive the final guarantee. As a result, they obtain $\mathcal{O}(1/(\mu\_x n)+d\_x\log(1/\delta)/(\mu\_x n^2\varepsilon^2))$ rate on the **primal risk** $\Phi(x) - \min\Phi(x)$ in near-linear time for SC-SC case and super-quadratic time $\mathcal{O}(n^{5/2})$ for SC-C case. However, our algorithms and analyses leverage the minimax structure and directly apply to the original problem. We obtain near-optimal guarantees on the stronger **duality gap** in near-linear time even for the C-C case.
>
> * **Generalization:** The population guarantees for DP-SMO in their work simply apply the generalization error of the primal function, but this is not always valid when the expectation and maximization **cannot be exchanged**. Instead, we use stability and generalization of the original minimax problem, measured by the duality gap.
>
> Finally, we want to emphasize again our key contributions on smooth convex-concave DP-SMO (see Table 1). Our results suggest that if the **nonsmooth** DP-SCO has some smooth convex-concave minimax reformulations, then **near-linear** time algorithms with near-optimal rate exist. We believe this makes an important initial step towards ultimately answering the long-standing open question on designing linear-time algorithms for nonsmooth DP-SCO.
>
>
> 2. **Gradient complexity in Theorem 3.4.**
>
> > A restriction on the smoothness parameter seems to be also required in order to be linear time. The units in the gradient complexity expression are not matching.
>
> We distinguish the following cases for the smoothness parameter $\ell$.
>
> * $\ell\leq\sqrt{n}$: Both Algorithm 2 and [21] obtain near-linear complexity.
>
> * $\sqrt{n}\leq\ell\leq n$: The stability analysis of SGD for smooth convex losses does not hold, and [21] cannot achieve any privacy guarantees.  However, our Algorithm 2 still provides the near-optimal rate, albeit with a super-linear complexity $n\leq T\leq n^{5/4}$ using Katyusha.
>
> * $\ell\geq n$: The smoothness does not help too much to improve the complexity. We can directly use algorithms for nonsmooth DP-SCO in [4, 29] with $\min(n^{5/4}d^{1/8}, n^{3/2}/d^{1/8}, n^2/\sqrt{d})$ complexity.
>
> Therefore, our relaxation is w.r.t. the regime $\sqrt{n}\leq\ell\leq n$ where previous algorithms for smooth DP-SCO **fail** to provide optimal guarantees, but we still achieve **near-optimal** rate with a better gradient complexity compared to the state-of-the-arts for nonsmooth DP-SCO. We will make this clear in our revised manuscript.
>
> The **unit-free** gradient complexity of SVRG is $\mathcal{O}(n + \sqrt{n}(\ell D/L))$ when plugging in $1/\mu\leq\sqrt{n}(D/L)$, where $L$ is the Lipschitz constant and $\Vert x^*\Vert\leq D$. The complexity of other algorithms can be derived similarly. The reason why we only keep $\ell$ in the complexity is to ease the comparison where $\ell$ can be ill-conditioned.  We will be precise about the above two in the revision.

---

### Official Review · Reviewer_mZUV · 2022-07-25

**Rating:** 8
**Confidence:** 2
**Soundness:** 4 excellent
**Presentation:** 4 excellent
**Contribution:** 3 good

**Summary:**

The paper proposes a framework for DP-SCO and DP-SMO, such that any non-private algorithm with a certain convergence guarantee can be plugged in. For smooth DP-SCO, it achieves the first near optimal guarantee without for algorithms apart from SGD; for smooth DP-SMO, it gives the first near linear time private algorithm.

**Questions:**

It all looks good to me.

**Strengths And Weaknesses:**

The paper provides the first near linear bound for DP-SMO.
The paper provides an algorithm-independent framework, thus extending the near linear time guarantee to algorithms other than DP-SGD.

---

> ### Author Response · Authors · 2022-08-02
> **Response to Reviewer mZUV**
>
> We thank the reviewer for the positive feedback.

---

### Author Response · Authors · 2022-08-02
**To All Reviewers and AC: Revision Update**

Thanks for the suggestions and we have updated the submission accordingly. All major changes are highlighted in **blue**. Here is a summary of the main updates in the revision:

1. **Missing related work:** See line 129-137 in related works, line 210-211 in Section 3.1, line 274-277 in Section 4.1 for discussions and comparisons.

2. **Lower-bound:** See Table 1 for a summary and line 268-270 in Section 4.1 for discussions of SC-SC case, line 323-324 in Section 4.2 of C-C case.

3. **Gradient complexity:** See line 225-228 in Remark 2 for unit-free complexity; line 236-241 for our restrictions w.r.t. $\ell$ compared to existing works.

4. **Imprecision:** We have modified the abstract and line 87-88, line 94 in contributions to be more precise about near-optimality. A footnote 1 is also included on page 2 for precise claims about "near-linear" and "near-optimal"; "Smooth" is added to each subsection head.

5. **Parameters $\mu$ and $L$:** We have added the parameter $\mu$ to input in Algorithm 1 and 3, and included some discussions of Lipschitzness $L$ in Appendix A.1, line 534-536.

**Others:** Change some typos, e.g., "convex" to "concave" in line 298; Rephrase some sentences to save space; Add paragraph headings in appendices for better structure, e.g. see line 580, 582; Rewrite Appendix B.4 for C-SC case in a more formalized way (see Corollary B.6.).

---

### Meta-Review · Area_Chair_NJdq · 2022-08-25

**Recommendation:** Accept
**Confidence:** Certain

**Metareview:**

This work studies minimax optimization for convex-concave objective. It studies the population loss version of this question and shows linear time differentially private algorithms for this problem that achieve the optimal privacy utility trade-off. The algorithm is based on the phased ERM approach.
The reviewers were in agreement that this problem is of interest and the paper makes a significant improvement on previous work to be interesting.  I would recommend acceptance.

**Award:**

No

---

### Decision · Program_Chairs · 2022-09-14

Accept